# Designing antifilarial drug trials using clinical trial simulators

Martin Walker [1,2 ✉], Jonathan I. D. Hamley[2,3,5], Philip Milton[2,3,5], Frédéric Monnot[4], Belén Pedrique[4] &
Maria-Gloria Basáñez[2,3]

Lymphatic filariasis and onchocerciasis are neglected tropical diseases (NTDs) targeted for elimination by mass (antifilarial) drug administration. These drugs are predominantly active against the microfilarial progeny of adult worms. New drugs or combinations are needed to improve patient therapy and to enhance the effectiveness of interventions in persistent hotspots of transmission. Several therapies and regimens are currently in (pre-)clinical testing. Clinical trial simulators (CTSs) project patient outcomes to inform the design of clinical trials but have not been widely applied to NTDs, where their resource-saving payoffs could be highly beneficial. We demonstrate the utility of CTSs using our individual-based onchocerciasis transmission model (EPIONCHO-IBM) that projects trial outcomes of a hypothetical macrofilaricidal drug. We identify key design decisions that influence the power of clinical trials, including participant eligibility criteria and post-treatment follow-up times for measuring infection indicators. We discuss how CTSs help to inform target product profiles.

[1] London Centre for Neglected Tropical Disease Research, Department of Pathobiology and Population Sciences, Royal Veterinary College, University of London, Hertfordshire AL9 7TA, UK. [2] London Centre for Neglected Tropical Disease Research, Department of Infectious Disease Epidemiology, Imperial College London, London W2 1PG, UK. [3] MRC Centre for Global Infectious Disease Analysis, Department of Infectious Disease Epidemiology, Imperial College London, London W2 1PG, UK. [4] Drugs for Neglected Diseases initiative, 15 Chemin Louis-Dunant 1202, Geneva, Switzerland. [5] These authors contributed equally: Jonathan I. D. Hamley, Philip Milton. ✉email: mwalker@rvc.ac.uk

The global health community aims to eliminate lymphatic filariasis (LF, elephantiasis) and onchocerciasis (river blindness) predominantly by mass drug administration (MDA) of antifilarial medicines[1,2]. The three cornerstone drugs to achieve this goal are albendazole either alone[3] or in combination with ivermectin and/or diethylcarbamazine (DEC) for LF[4], and ivermectin alone for onchocerciasis. In 2017, 465.4 million people received treatment for LF and 142.4 million people received ivermectin for onchocerciasis[5,6]. Ivermectin and DEC have potent activity against the microfilarial progeny (the stage transmitted to vectors) of adult worms (macrofilariae). For onchocerciasis, ivermectin also temporarily inhibits the production of microfilariae (mf) by adult female *Onchocerca volvulus*[7] and is partially macrofilaricidal after multiple doses[8]. For LF, combinations of DEC and albendazole[9], and triple combination therapy, IDA (ivermectin with DEC and albendazole)[4], has significant activity against adult worms.

The suppression of mf elicited by antifilarial drugs—combined with their efficacy in single oral doses—makes them perfectly suited for lowering and potentially interrupting transmission of filarial infections by MDA if delivered at a high coverage and treatment adherence, and on either an annual or semi-annual basis[2]. For LF, the microfilaricidal (and macrofilaricidal) efficacy of the drugs, and the shorter life-expectancy of the adult worm compared to *O. volvulus* (about 5 years compared to about 10 years), has led to the expectation that widespread elimination is feasible, particularly if MDA is combined with vector control[10,11]. As of 2017, 21 out of 73 LF-endemic countries had stopped MDA and transitioned to post-treatment surveillance[6]. For onchocerciasis, MDA has greatly reduced morbidity and excess mortality[12,13] and has successfully eliminated onchocerciasis transmission from Colombia[14], Ecuador[15], northern Venezuela[16], Mexico[17], and Guatemala in Latin America[18]. Good progress towards elimination has also been made in Africa[19], which bears 99% of the onchocerciasis cases, with notable successes in foci in Mali, Senegal[20,21] Nigeria[19] and Sudan[22].

However, the long-lived nature of adult *O. volvulus*, and the persistence of focal areas of intense transmission mean that elimination of onchocerciasis at a country scale will be extremely challenging within currently proposed timeframes[23]. Despite many years of MDA, transmission continues in highly endemic onchocerciasis communities with high vector biting rates[24–27]. Moreover, in areas of Central and West Africa where loiasis (African eye worm[28], caused by another filarial parasite, *Loa loa*) occurs co-endemically with onchocerciasis, ivermectin cannot be safely delivered through routine MDA because of the rare but severe and life-threatening reactions associated with the killing of *L. loa* mf in heavily microfilaraemic individuals[29]. Additionally, suboptimal responses to ivermectin (potentially indicating loss of drug efficacy) have been reported phenotypically in Ghana[30] and confirmed genetically in Ghana and Cameroon[31]. Hence, there is a growing consensus that it is unlikely that ivermectin alone will be sufficient to eliminate onchocerciasis in all endemic African settings and that drugs with macrofilaricidal properties are pressingly needed[32–34]. Macrofilaricides will help accelerate progress towards country-wide elimination and improve individual patient treatment options. Public and privately funded drug development partnerships for the neglected tropical diseases (NTDs), including onchocerciasis, have been developed in response to this need[35–38].

Clinical trial simulation is the (mathematical) representation of clinical trials to inform decision-making on trial design by forward projecting likely trial outcomes[39], e.g., Vegvari et al.[40] Simulation has been widely implemented in the pharmaceutical sector to assist with the design of trials, often seeking to balance the cost of collecting data with the information (such as safety and/or efficacy) on the drug under consideration that these data will provide[41]. Clinical trial simulation has not been used to assist drug development in the NTD domain, although simulation has been used widely to project the impact of population-level interventions on onchocerciasis, LF and other NTDs[42], and to inform the design of cluster intervention trials targeting elimination of soil-transmitted helminthiases[43,44]. Simulation could inform the design of clinical trials, offering resource savings that may be particularly important for NTD drug development.

Simulators can include all aspects of the trial protocol, from the recruitment of participants meeting pre-defined eligibility criteria in simulated populations, to the projection of trial outcomes under desired drug properties defined by a target product profile (TPP). Simulations can answer questions such as: (a) how many trial participants need to be recruited to demonstrate superiority over existing treatments (in two- or multi-armed trials); (b) when should participants be followed up, and (c) what infection indicators should be measured as primary and secondary outcomes? These questions are particularly pertinent to the design of antifilarial drug trials because: (a) drugs are generally not completely curative, yet can exert long-lasting reductions in infection intensities; (b) macrofilaricidal drug responses are typically quantified indirectly by measuring mf; (c) response variation among participants is generally very high, and (d) reinfection by unexposed drug-naïve parasites during trials conducted in endemic settings is inevitable (and can be accounted for using trial simulators that explicitly model transmission).

Here we illustrate how a clinical trial simulator (CTS) can be used to help design antifilarial drug trials, using as an example a hypothetical macrofilaricidal drug for the treatment of onchocerciasis. We use an adaptation of our individual-based onchocerciasis transmission model, EPIONCHO-IBM[45,46] (a stochastic analogue of the well-established EPIONCHO transmission model)[47–49], to simulate a hypothetical phase IIb two-arm clinical trial (i.e. a trial focused on assessing efficacy but more limited in size than a phase III trial) comparing the efficacy of the hypothetical macrofilaricide to ivermectin. The CTS can model the (modifiable) antifilarial action of macrofilaricides and factors defining participant eligibility. We focus on trials conducted in previously ivermectin-naïve, mesoendemic transmission foci with a microfilarial prevalence among individuals aged ≥ 5 years ranging from 40% to 50% (transmission foci with higher microfilarial prevalence are likely to have been undergoing MDA for many years). We compare drug responses elicited by a macrofilaricide that is either purely macrofilaricidal (macrofilaricidal only macrofilaricide, MOM) or that has accompanying microfilaricidal activity (macrofilaricidal and microfilaricidal macrofilaricide, MAMM), and identify opportune sampling (follow-up) times and associated sample sizes for demonstrating superiority of these drugs (MOM and MAMM) compared to ivermectin-treated control groups. We explore how design choices related to participant eligibility criteria and parasitological sampling affect the efficiency of trials and we discuss our results in the context of the definition and refinement of TPPs, the need for improved diagnostic indicators (biomarkers) of patent onchocerciasis, and the implementation of trials within a landscape of MDA.

## Results

**Clinical trial simulation.** A schematic description of the simulated trial design is given in Fig. 1 and of the individual-based onchocerciasis transmission model, EPIONCHO-IBM in Fig. 2. An overview of the varied eligibility criteria, diagnostic protocols, macrofilaricidal efficacies and assumed pharmacodynamics (PD) properties of the hypothetical MOM/MAMM is given in Table 1. Macrofilaricidal effects are modelled as the percentage of adult *O.*

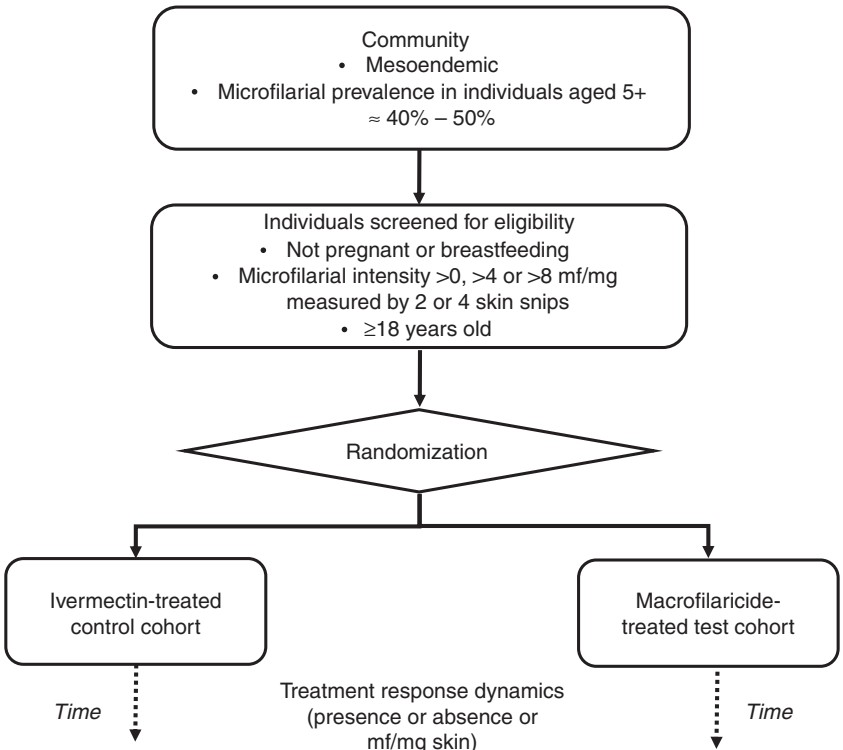

**Fig. 1 Schematic representation of the simulated trial design.** The simulated trial is conducted in a community previously ivermectin-naïve residing in a lower-end mesoendemic setting with a microfilarial prevalence among individuals aged ≥5 years of 40% to 50%. It is assumed that all individuals are screened for eligibility to participate based on their age (≥18 years), their pregnancy and breastfeeding status, and their infection status (presence/absence of microfilariae, mf) and microfilarial load (mf/mg) in the skin based on taking 2 or 4 skin snips. The presence and intensity (density) of mf in the skin, age, sex, and pregnancy and breastfeeding status of each individual are modelled using an adapted version of EPIONCHO-IBM, along with individuals' parasitological response to treatment with ivermectin or a hypothetical macrofilaricide that is macrofilaricidal only (MOM) or both macrofilaricidal and microfilaricidal (MAMM).

*volvulus* killed within three months of treatment. The micro-filaricidal effect of a MAMM is assumed to be identical to that of ivermectin (but without the temporary sterilisation or so-called embryostatic effect, see Basáñez et al.[7]). Mathematical details of EPIONCHO-IBM, including a complete description of the PD properties of ivermectin, macrofilaricides and other adaptions implemented to yield the CTS, are given in the Supplementary Methods.

**Treatment response dynamics**. The different PD properties of hypothetical macrofilaricides elicit profoundly different dynamics in microfilarial outcome measures (the arithmetic mean number of mf/mg of skin, i.e., microfilarial intensity, or the percentage of participants positive for mf, i.e., microfilarial prevalence) with time after treatment (Fig. 3 and Supplementary Fig. 1 for an upper-end mesoendemic setting with a pre-treatment micro-filarial prevalence among individuals ≥5 years of 50%). A MOM elicits a slow and sustained decline in mf caused by the decreased rate of replenishment of mf following the death of adult worms. The rate of attrition is driven by the approximate 10-month average life span of mf and the nadir (the lowest density of microfilaridermia following treatment) is determined by the competing effects of macrofilaricidal activity and reinfection by drug-naïve worms. Incomplete clearance of adult parasites and ongoing reinfection result in the mf population not declining to 0 and gradually repopulating (Fig. 3). The microfilaricidal activity of a MAMM reduces the microfilarial population more rapidly, followed by a similarly sustained suppression of mf. The more transient effects of ivermectin arise because female worms regain fertility and resume production of mf[7], effecting a more rapid

'bounce back' in the parasite population. There is negligible qualitative difference in the treatment response dynamics between the lower and upper end of the mesoendemicity setting (compare Fig. 3 with Supplementary Fig. 1).

**Opportune follow-up timeframes**. The response dynamics define opportune follow-up timeframes for demonstrating a difference between outcome measures among macrofilaricide-treated (test) and ivermectin-treated (control) groups. Defining the optimal follow-up times is a balance between the average response among participants in different groups (i.e., when the difference between the means is greatest, Fig. 4 and Supplementary Fig. 2 for an upper-end mesoendemic setting) and the variability in responses among participants within the same group (i.e., when the estimated difference between means is suitably precise, Fig. 4). For example, the greatest difference in the average microfilarial intensity between treated (with either MOM or MAMM) and control groups occurs between 30 and 36 months after treatment. However, the uncertainty associated with this difference is also greatest at these long follow-up times (Fig. 4a, b). At follow-up times less than 12 months after treatment, ivermectin is observed to be either superior or equally efficacious as the macrofilaricides (MOM/MAMM) because of its rapid microfilaricidal activity (and concomitant embryostatic effect). The MOM begins to elicit a superior response compared to ivermectin between 12 and 24 months after treatment (depending on the macrofilaricidal efficacy, Fig. 4a, c). The MAMM yields superior reductions in mf compared to ivermectin sooner, between 6 and 12 months (depending on macrofilaricidal efficacy), because of its accompanying microfilaricidal activity

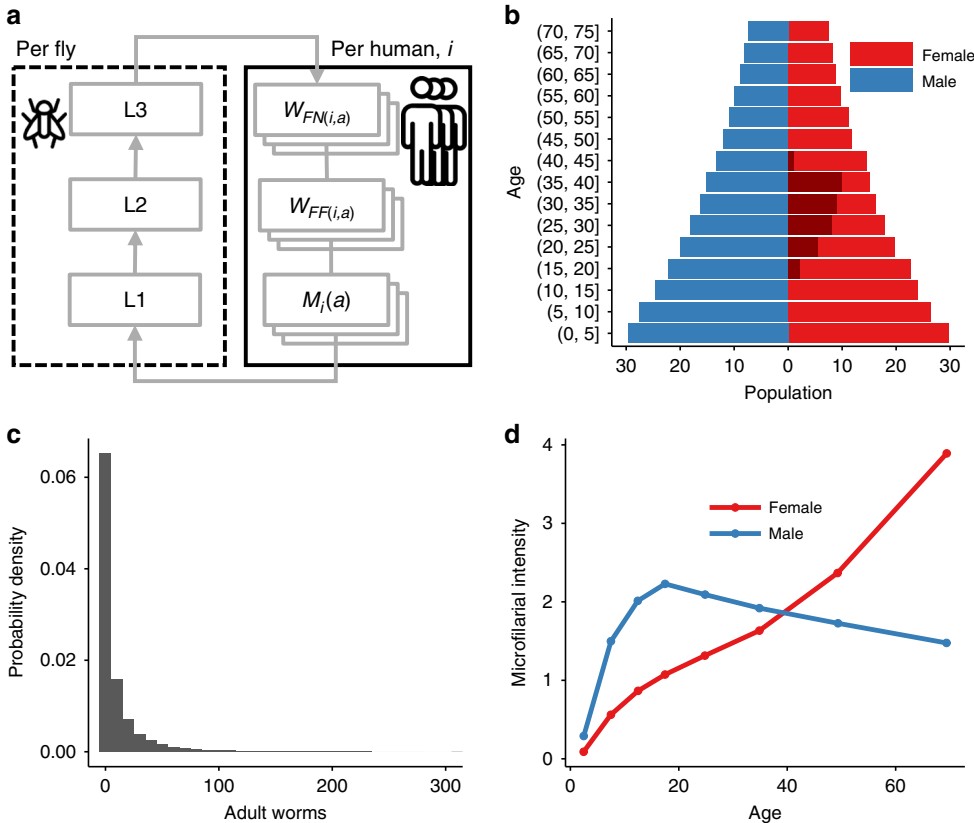

**Fig. 2 Schematic description of EPIONCHO-IBM.** EPIONCHO-IBM is an individual-based analogue of the population-based (deterministic) EPIONCHO transmission model[47, 49] tracking the number and age ($a$) of adult non-fertile ($W_{FN(i,a)}$) and fertile ($W_{FF(i,a)}$) female *Onchocerca volvulus* worms, microfilariae ($M_i(a)$) and male worms (not represented schematically) within individual (human) hosts ($i$) and the mean number of L1, L2 and L3 larvae in blackfly vectors (**a**). Host births, deaths and age are based on the typical demography of rural low-income communities in Africa (**b**) and we additionally model explicitly the pregnancy and breastfeeding status of women between the ages of 16 and 40 years (shaded dark red (**b**)). Individuals are differentially exposed to blackfly bites, driving the typically overdispersed (aggregated) distribution of parasites among hosts, such that most people are either uninfected or lightly infected and few have many adult parasites (and microfilariae) (**c**). Exposure to blackfly bites depends on age and varies between boys and girls/ men and women, and is implicitly related to the amount of time spent in areas of high blackfly (vector) density. In this parameterisation, based on data from an onchocerciasis focus in northern Cameroon, girls are relatively less exposed than boys and women are relatively more exposed than men (**d**)[78]. The parasite population is regulated by density-dependent processes operating within the blackfly vector, and on the establishment of incoming worms and the mating of (female) adult worms with the human host. Female worms in a host are assumed to produce microfilariae if at least one male worm is present (i.e., polygamous mating).

---

**Table 1 Parameters varied in the clinical trial simulation.**

| Parameter | Values |
| --- | --- |
| Endemicity | 40% or 50% microfilarial prevalence among individuals aged ≥5 years |
| Inclusion criteria | >0, >4 or >8 microfilariae /mg skin |
| Diagnostic protocol | 2 or 4 skin snips per participant |
| Macrofilaricidal efficacy[a] | 60%, 75% or 90% |
| Pharmacodynamics (PD) profile | Macrofilaricidal-only macrofilaricide (MOM) or macrofilaricidal & microfilaricidal[b] macrofilaricide (MAMM) |

[a]Macrofilaricidal efficacy is defined as the probability (expressed as a percentage) that an adult *Onchocerca volvulus* worm is killed within three months of (single dose) treatment with the hypothetical drug.
[b]Microfilaricidal effect is assumed to be identical to that of ivermectin (excluding the temporary sterilisation or so-called embryostatic effect, see Supplementary Methods).

---

(Fig. 4b, d). Inference on opportune follow-up timeframes is qualitatively very similar between the lower and upper end of the mesoendemicity setting, albeit there is noticeably greater variation in the difference between the microfilarial intensity outcome measure at the upper endemicity end of the setting (compare Supplementary Fig. S2a, b with Fig. 4a, b).

**Estimated sample sizes.** The sample size required to achieve a given statistical power to detect a difference between (ivermectin-

treated) control and (macrofilaricide-treated) test groups will depend on the difference in the average response between groups (effect size), the variability associated with this difference and on the chosen outcome measure (microfilarial intensity or pre-valence). For each outcome measure, effect size and variability depend on the PD properties of the macrofilaricide (MOM/ MAMM), the follow-up timeframe and the macrofilaricidal efficacy (Fig. 4). The CTS approach permits exploration of how decisions on minimum infection inclusion criterion and the

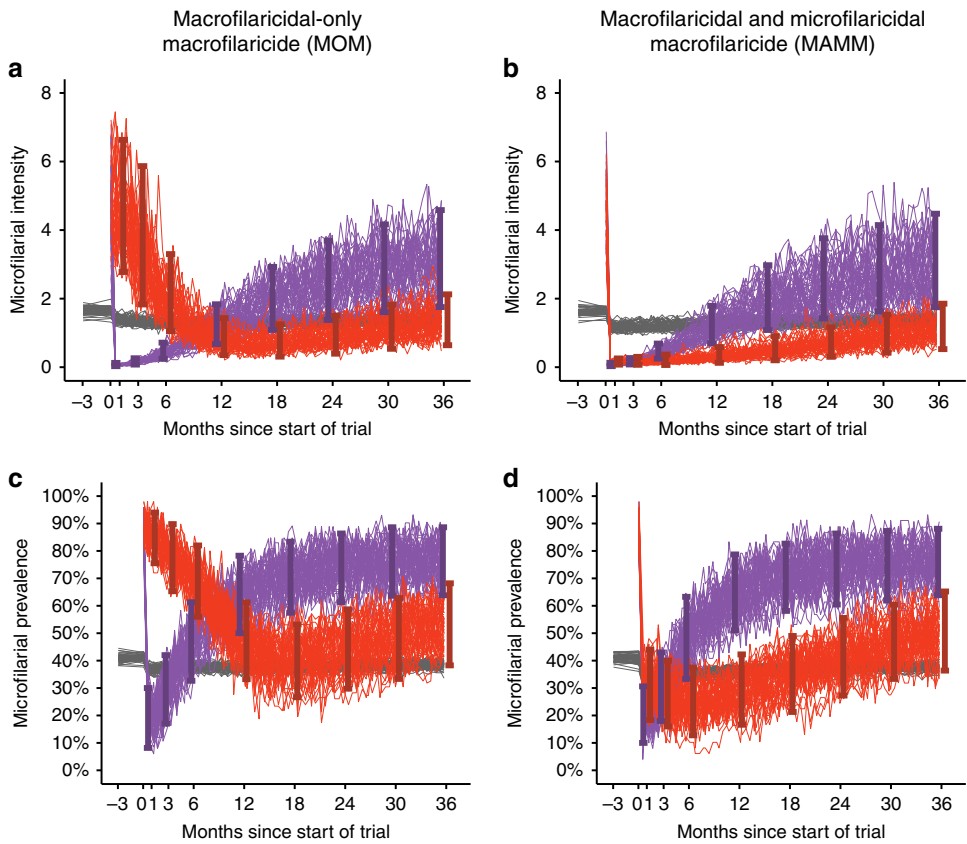

**Fig. 3 Parasitological response dynamics in ivermectin-treated and macrofilaricide-treated participants.** Each panel depicts the parasitological response dynamics in two 50-participant cohorts treated with either ivermectin (purple lines) or a hypothetical macrofilaricide (red lines) in an ivermectin-naïve lower-end mesoendemic setting. The macrofilaricide has an efficacy of 90% (90% of adult *Onchocerca volvulus* are killed within three months of treatment) and either macrofilaricidal activity only (MOM, left-hand side (**a**, **c**)) or macrofilaricidal and microfilaricidal activity (MAMM, right-hand side (**b**, **d**)). The parasitological outcome measure is either the arithmetic mean number of microfilariae (mf) per mg of skin (microfilarial intensity (**a**, **b**)) or the percentage of participants positive for mf (microfilarial prevalence (**c**, **d**)), both measured by 2 skin snips. Participants were included in the cohort if they were positive for mf (i.e., inclusion criterion was >0 mf/mg of skin). Each thin line represents a single simulation and the vertical error bars indicate the range which captures 95% of the simulations. The grey lines indicate the mean (either mf per mg of skin, i.e., intensity, or presence of mf, i.e., prevalence) in the whole population, which is assumed to comprise 1000 individuals. Note that only eligible participants are treated, i.e., (community-wide) mass drug administration is not simulated.

number of skin snips used to detect mf (Table 1 and Fig. 1) also influence the required sample size. For examples of trials employing different design decisions see Awadzi et al.[50], Batsa Debrah et al.[51], Opoku et al.[52], and Turner et al.[53].

Estimated sample sizes required to detect with 80% power a statistically significant superior response (using either microfilarial intensity and prevalence) in the macrofilaricide-treated test group compared to the ivermectin-treated control group at various times after treatment, macrofilaricidal efficacies, inclusion criteria and number of skin snips are shown in Fig. 5. Sample sizes decrease with increasing macrofilaricidal efficacy, and marginally for microfilarial intensity (because of the increased measurement precision) for 4 versus 2 skin snips. More pronounced effects result from the choice of outcome measure and infection level inclusion criteria. Sample sizes are generally lower when using microfilarial prevalence compared to microfilarial intensity for an inclusion criterion of >0 mf/mg because of the lower inter-participant variability associated with the former. However, for an increasing minimum microfilarial intensity inclusion criterion (ranging from >0 mf/mg to >8 mf/mg) opposing directional effects on sample sizes for the different outcome measures are evident. For microfilarial intensity, increasing the infection level inclusion criterion decreases required sample sizes because inter-participant variability in

microfilarial counts is reduced (by selecting individuals with somewhat more similar microfilarial loads). By contrast, for microfilarial prevalence, increasing the infection level inclusion criterion increases the sample size. This is because the selection of more heavily infected participants decreases the number of participants who are ostensibly 'cured' (i.e., achieving zero mf) following treatment (note that macrofilaricidal efficacy is defined probabilistically as the chance that an adult *Onchocerca volvulus* worm is killed by treatment). These contrasting effects are enhanced in the upper-end mesoendemic setting because participants tend to be more heavily infected (Supplementary Fig. 3).

## Discussion

We have illustrated, using onchocerciasis as an example, how a CTS can be used to inform the design of antifilarial drug trials by projecting responses in outcomes measured in test and control groups (cohorts) receiving either a (hypothetical) macrofilaricidal treatment or an existing (predominantly microfilaricidal) comparator therapy. The transmission model that underpins the CTS accounts for the inevitable reinfection of participants by drug-naïve parasites in endemic settings. Crucially, this permits a priori identification of opportune follow-up times after treatment when the balance between measurable drug effects and reinfection is most favourable. Moreover, the individual-based structure of the

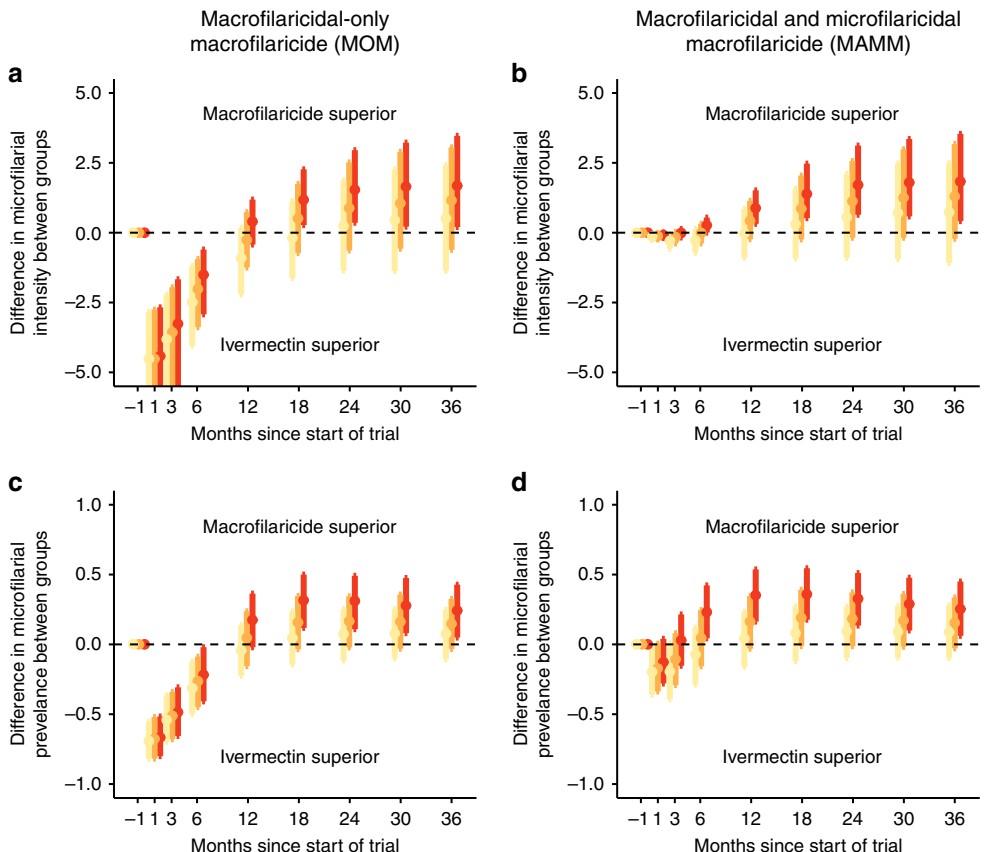

**Fig. 4 Difference in parasitological outcomes between ivermectin-treated and macrofilaricide-treated participants.** Each panel depicts differences in parasitological outcome measures between two 50-participant cohorts treated with either ivermectin (control group) or a hypothetical macrofilaricide (test group) in an ivermectin-naïve lower-end mesoendemic setting. The macrofilaricide has an assumed efficacy of either 60% (cream points and lines), 75% (light orange points and lines) or 90% (red points and lines). Efficacy refers to the percentage of adult *Onchocerca volvulus* killed within three months of treatment. Left-hand side panels (**a**) and (**c**) show results for a macrofilaricidal only macrofilaricide (MOM); right-hand side panels (**b**) and (**d**) show results for a macrofilaricidal and microfilaricidal macrofilaricide (MAMM). The parasitological outcome measures compared are either the difference in arithmetic mean number of microfilariae (mf) per mg of skin (microfilarial intensity, panels (**a**) and (**b**)) or the difference in percentage of participants positive for mf (microfilarial prevalence, panels (**c**) and (**d**)), both measured by 2 skin snips. Participants were included in the cohort if they were positive for mf (i.e., infection status inclusion criterion was >0 mf/mg of skin). Points indicate the arithmetic mean of the simulated differences in the average response between treatment groups and vertical error bars indicate the range which captures 95% of the differences. The horizontal dashed line indicates the threshold of equivalence (i.e., difference = 0), above or below which the MOM/MAMM or ivermectin exhibit superiority, respectively.

transmission model permits explicit simulation (and exploration) of other proposed protocols, such as the use of minimum infection level eligibility criteria and sampling procedures (e.g., parasitological sampling by skin snips). Explicit simulation of transmission and trial protocols represent the key benefits of using a CTS to inform trial design.

For the PD properties of macrofilaricides (MOM/MAMM) considered here, the best follow-up times (requiring the smallest sample sizes) occur at least (but often greater than) 12 and 18 months after treatment. But follow-up times and required sample sizes are also strongly influenced by the presumed efficacy of the treatment, such that for the lowest 60% efficacy considered here, follow-up times and/or required sample sizes may be deemed too long to be feasible. Hence, the CTS plays an important role in evaluating and revising a TPP as being demonstrable within a clinical trial framework. The CTS also permits the impact of other frequently considered trial design choices—such as pre-selection of individuals with a minimum intensity of infection[6,40], protocols to increase diagnostic sensitivity (i.e., 4 versus 2 skins snips)[54] and the choice of primary outcome measure (i.e., prevalence versus intensity)—to be evaluated in practically relevant quantities such as required sample sizes.

Long follow-up times of 1 year to 18 months are a feature of antifilarial drug trials (not necessarily including macrofilaricidal therapies) because of the protracted and often non-curative parasitological response to treatment[53,55–57]. Trials of MOMs compared to predominantly microfilaricidal drugs incur particularly protracted follow-up times because of the indirect nature of parasitological outcome assessment; mf populations decrease by natural attrition after the removal of reproductively active female worms[45]. This is also a feature of microfilarial responses in clinical trials of anti-*Wolbachia* drugs that deplete the endosymbionts of *O. volvulus* and lymphatic filariae causing female worm sterility and eventual death[58,59]. The heavy reliance on mf as a primary outcome measure is because they are the most accessible parasite stage and, in onchocerciasis, because there is not any other reliable indicator of patent infection (e.g. ultrasound examination of onchocercomata for signs of adult worm viability)[60]. Serology[61] is not suitable as it predominantly tests for past exposure to *O. volvulus* antigens[62]. This is unlike LF, for which (in addition to ultrasonography)[63] there exist circulating filarial antigen (CFA) tests[64,65] which can be used as alternatives to sampling mf[66,67] and offer a more direct means of detecting macrofilariae.

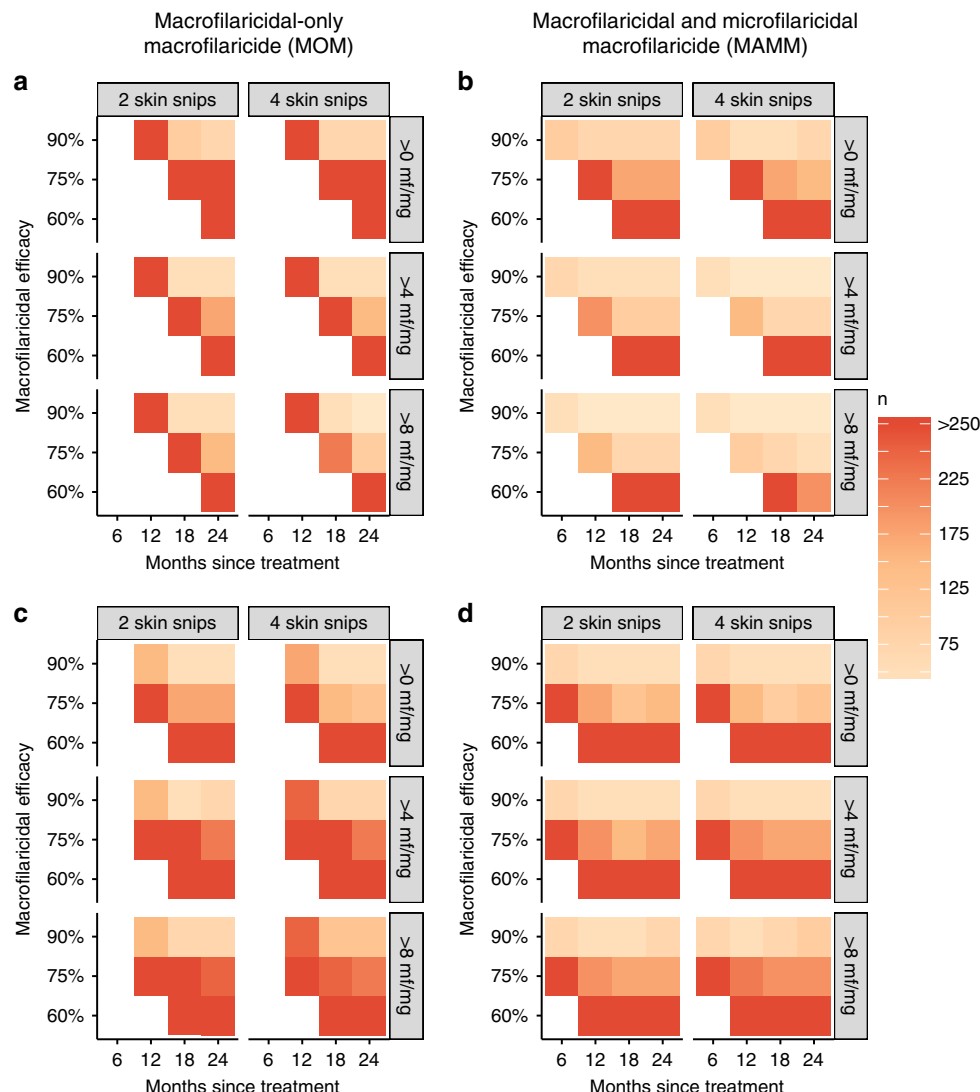

**Fig. 5 Sample sizes required to detect a superior response in macrofilaricide-treated compared to ivermectin-treated participants.** Each panel depicts sample sizes ($n$) required to detect, with 80% power, a statistically significant superior response in the macrofilaricide-treated test group compared to the ivermectin-treated control group in an ivermectin-naïve lower-end mesoendemic setting. Sample sizes are calculated at 6, 12, 18 and 24 months after treatment assuming a 10% loss to follow up per year. Macrofilaricidal efficacy (either 60%, 75% or 90%) corresponds to the percentage of adult *Onchocerca volvulus* killed within three months of treatment with a macrofilaricidal only macrofilaricide (MOM, left-hand side; panels (**a**) and (**c**)) or a macrofilaricidal and microfilaricidal macrofilaricide (MAMM, right-hand side; panels (**b**) and (**d**)). The parasitological outcome measures are the arithmetic mean number of microfilariae (mf) per mg of skin (microfilarial intensity, panels (**a**) and (**b**)) and the percentage of participants positive for mf (microfilarial positive, panels (**c**) and (**d**)), both measured by 2 or 4 skin snips and using an infection eligibility criterion before treatment of either >0 (i.e., presence of mf), >4 or >8 mf/mg of skin. Empty squares correspond to time points when the response in the macrofilaricidal-treated test group is inferior to ivermectin.

It is possible to make direct observations on the fertility and viability of adult *O. volvulus* extracted from superficial (palpable and accessible) onchocercomas (nodules) (see for example Gardon et al.)[68]. For directly acting macrofilaricides, this could be used to assess efficacy earlier than is apparent in microfilarial outcome measures. It also permits better understanding of drug activity against adult filariae. Direct assessment of adult worms collected by nodulectomy has been used extensively to demonstrate the efficacy of anti-*Wolbachia* therapies in depleting endosymbiotic *Wolbachia*[58,69], albeit the indirect killing effect of these therapies means that, even by this approach, detecting macrofilaricidal activity still requires follow-up times in excess of 18 months[59]. However, although these methods are useful as secondary outcome measures, it remains difficult to design and power trials using adult worms as the primary outcome measure.

Infection with adult worms is typically diagnosed by identifying palpable nodules but the relationship between the presence of nodules and numbers of adult worms is highly uncertain[70], and there do not exist validated statistical models that describe the number of adult worms typically found in onchocercomata (but see Duerr et al.)[71]. Ideally, an *O. volvulus* antigen test would become available to replace the invasive procedures of nodulectomy or skin snipping[72].

The variability in estimated sample sizes required to detect superiority of a macrofilaricide compared to ivermectin reflects the potential resource-saving benefits and efficiency gains that clinical trial simulation can offer. For example, even to demonstrate superiority of a macrofilaricide with a high 90% target efficacy, estimates range from ≤25 to >250 participants per cohort depending on the chosen outcome measure (microfilarial

intensity or prevalence), the follow-up time and the inclusion criterion for minimum infection level. For drugs with the lowest assumed macrofilaricidal efficacy of 60%, the trial design can make the difference between being able or unable to demonstrate superiority compared to ivermectin (i.e., to make the go/no go decisions). For example, our results suggest that it is not possible to demonstrate superiority of a MOM with 60% efficacy with a follow up shorter than an 18-month and even at long (18- to 24-month) follow ups, the required sample sizes will exceed 250 per treatment group). It is a noteworthy limitation that in this CTS, variation among drug responses is driven only by stochastic variation in numbers of adult worms and numbers (or presence) of mf detected by 2 or 4 skins snips (see Supplementary Methods). No consideration is given to inter-participant variation in treatment responses[73] (PD variation) which would inflate sample sizes.

The comparatively large required sample sizes and long follow-up times for the hypothetical MOM represents a general challenge to the evaluation of novel or repositioned antifilarial drugs within a clinical trial structure. It is conceivable that sustained suppression of mf elicited by drugs with even only moderate (60%) macrofilaricidal activity is at least as beneficial to patients than the rapid but transient clearance of mf elicited by many microfilaricides (e.g. ivermectin and DEC). To demonstrate this with clinical outcome measures (such as the prevalence of pruritus or troublesome itch, other skin pathologies and ocular pathologies) that typically lag behind parasitological indicators of infection[74], would require prohibitively long follow-up periods that would be difficult to justify ethically when the *de facto* standard of treatment for people living in endemic areas is annual MDA. Indeed, on this basis, proposed trials with follow-up times greater than 12 months may face ethical challenges and require that ivermectin (for onchocerciasis) or ivermectin with albendazole or DEC (for LF) be offered to participants 1 year after treatment or as 'rescue' medication if infection is detected.

A seemingly natural solution to this problem is to design trials that concomitantly deliver the standard annual antifilarial therapy and compare outcomes in groups treated (not necessarily at the same time) with a macrofilaricide and a standard MDA treatment or treated with standard MDA alone. This type of strategy has been used in trials of anti-*Wolbachia* therapies for onchocerciasis and LF which elicit a similarly protracted decline in mf (caused by female worm sterilisation and eventual death following depletion of endosymbiotic *Wolbachia*), when used as monotherapy[53], to the hypothetical MOM considered here, but which effect lasting suppression and lower rates of microfilarial repopulation after subsequent rounds of ivermectin treatment. Whether a MOM would be given at the same time as ivermectin, or with sufficient time before or after MDA rounds, would depend on the safety of co-administration and on whether or not it would be desirable to have a MOM registered exclusively as a component of a combination therapy with ivermectin.

Trials combining a MOM and ivermectin (whether or not as a combination therapy), however, may be challenging in areas co-endemic with the filarial parasite *Loa loa* because the killing of *L. loa* mf by microfilaricides in heavily microfilaraemic individuals can lead to severe and sometimes fatal severe adverse events (SAEs)[29]. Indeed, the disruption to onchocerciasis and LF control caused by SAEs in communities co-endemic with loiasis is a key driver behind the current push to develop safe antifilarial drugs for loiasis-infected patients[32] and test-and(not)-treat strategies[75]. In many such communities, individuals with high intensities of *L. loa* microfilaraemia must be identified and excluded from receiving microfilaricidal drugs (i.e., by test-and-not-treat strategies)[75]. Hence, there remains a potential use case for MOM monotherapy (assuming a MOM does not affect *L. loa* mf) or indeed anti-

*Wolbachia* therapies (that do not affect *L. loa* because they do not harbour *Wolbachia* endosymbionts) in onchocerciasis-loiasis co-endemic communities.

An alternative approach to earlier identification of macrofilaricidal activity is to collect interim parasitological (microfilarial) data before the indicated optimal follow-up time. While these data may not indicate superiority compared to ivermectin, model-based interpretation of the data (for example by fitting the CTS or related transmission dynamics models)[8,59] will give an indication of the underlying activity and efficacy of the trialled drug. This may give important early indications to support 'go / no go' decisions and/or inform refinement or adaption of the trial design. For example, measuring mf from patients at interim time points—before the initially-proposed most opportune follow-up time—would permit the CTS to be fitted to the data to achieve an early indication/estimate of macrofilaricidal activity and accompanying microfilaricidal activity. Decisions could then be made on whether to continue the trial (e.g., depending on how the early estimate of macrofilaricidal activity compares to the desired TPP) and when the final follow up should be made (e.g., depending on the indicated level of microfilaricidal activity). The efficiency gains associated with such adaptive (as opposed to 'fixed') trial designs is well recognised and has led to their increased use across the clinical medicine domain[76,77]. Collection of interim longitudinal repeated measurements also confers added statistical power to post hoc analyses and can supplement the final comparison of trialled treatments.

Conducting antifilarial drug trials in the current global intervention landscape of onchocerciasis and LF, where many moderate to high intensity endemic communities have been undergoing MDA for many years, poses significant logistical and inferential challenges to trial design. For example, the use of a high minimum infection inclusion criterion will make it harder to find sufficient numbers of eligible participants (see Supplementary Fig. 4) within single communities or transmission foci with similar intensities of transmission (driven by local density of vectors). Hence, if choosing microfilarial intensity as the desired outcome measure, there is a trade-off between the benefit of reducing variability in treatment responses (and lower sample sizes) by selecting participants with more similar (and higher) infection intensities and the challenge of recruiting sufficient numbers. Moreover, in communities where awareness of onchocerciasis has diminished—following years of MDA that has effectively controlled the disease as a public health problem—the challenges of recruiting and retaining participants during the trial may increase.

Designing trials in communities with long histories of MDA adds further complexity and additional uncertainty. For example, even given at annual MDA frequencies, ivermectin exerts notable (but uncertain) macrofilariacidal effects on *O. volvulus* worms and cumulative reductions in female worm fertility[8] which will propagate into the projected response dynamics following treatment. Hence, assumptions on the cumulative effect on female *O. volvulus* of multiple exposures to ivermectin will likely have a substantial effect on projected response dynamics. Moreover, obtaining accurate and reliable information on how many prior rounds of treatment individual participants have received is a challenge, although information on how many rounds of MDA have been distributed at a population level is more readily available. Hence, even if trials cannot be conducted in treatment-naïve communities, we suggest that efforts should be made to select participants with as few as possible previous antifilarial treatments.

Notwithstanding these challenges, the CTS presented here can in principle be used to simulate trials conducted in a variety of epidemiological settings, with different histories of MDA.

Parameters relating to the core transmission dynamics of onchocerciasis have been estimated and refined in numerous previous studies[45,47,78,79] and capture well the observed epidemiological relationship between ABR and microfilarial prevalence and intensity[45]. Consequently, the CTS can reflect transmission conditions across the wide spectrum of endemic settings by varying the annual biting rate (ABR) of blackfly vectors. It can also be used, more cautiously, to simulate trials conducted in a backdrop of MDA by varying parameters that control the number, frequency and coverage of past and ongoing mass treatments with ivermectin[45]. Installation and use of the CTS and examples of simulating trials in a variety of epidemiological contexts are given in the README.html file at https://mrc-ide.github.io/EPIONCHO-IBM-CTS/README.html. The README.html also includes instructions on how to vary parameters relevant to the trial design, such as those varied in this work (Table 1), and additional parameters, such as the participant dropout rate (which was here set to a nominal 10% per year).

We have illustrated how a CTS can be used to inform the design of antifilarial drugs, using as an example a simple two-arm (phase IIb) trial of a hypothetical macrofilaricide compared to ivermectin for the treatment of onchocerciasis. The approach helps to maximise the efficiency of trials, indicating opportune follow-up times for detecting desired drug properties and aiding decision making on TPPs, participant eligibility criteria and sampling protocols. The CTS developed here could be extended to simulate more complex trials, including trials with multiple comparator arms, combinations of therapies either given concurrently or intermittently to meet existing standards of regular MDA-based treatment (in addition to the existing capability to project responses to treatment in communities with past and ongoing MDA). Because the CTS presented here is based on a full transmission model of infection, it is readily extendable to project the potential wider epidemiological impact of new antifilarial drugs or combinations, an approach that has been used to support the registration of moxidectin for the treatment of onchocerciasis[80] and the recent recommendation by the World Health Organization (WHO) to endorse the IDA (ivermectin, DEC and albendazole) triple therapy for LF[81]. We believe that the resource-saving payoffs of prospective trial simulation make them an essential tool for designing drug trials in the resource-limited context of filarial infections and other NTDs.

## Methods

**Model overview.** We developed a CTS from the individual-based onchocerciasis transmission model EPIONCHO-IBM[45] to simulate a trial conducted in an ivermectin-naïve mesoendemic community with a microfilarial prevalence in those aged ≥5 years of either 40% or 50%. A full mathematical description of the model is given in Hamley et al[45]. and also in the Supplementary Methods. Briefly, EPIONCHO-IBM is an analogue of the population-based EPIONCHO model[47,49] tracking the number of adult *Onchocerca volvulus* worms of both sexes and microfilariae (mf) within individual (human) hosts (Fig. 2a). Host births and deaths are based on the typical demography of rural low-income communities in Africa (Fig. 2b) and individuals are differentially exposed to blackfly bites, driving in a mechanistic fashion the typically overdispersed (aggregated) distribution of parasites among hosts (Fig. 2c). Exposure to blackfly bites varies with age and between males and females, following epidemiological patterns typically found in West Africa[78] (Fig. 2d). Sexual reproduction of *O. volvulus* is modelled by assuming that only female worms in hosts concurrently infected with at least one male worm will produce mf. The age, sex, pregnancy and breastfeeding status (assuming that women between 16 and 40 may be pregnant or breastfeeding for 2 years; i.e., after a 9-month pregnancy and 1 year and 3 months of breastfeeding) of each human in the population is modelled (Fig. 2b).

**Participant eligibility and randomisation.** We assumed that everyone aged ≥18 years in a population of 1000 individuals who were not pregnant or breastfeeding were screened for mf (by skin snipping). We further assumed that a set (target) number of eligible individuals consented to participate (we targeted 50 participants per cohort for the purpose of approximating the mean and variance among individual response dynamics to estimate sample sizes). We randomly assigned

consenting participants to receiving either a hypothetical macrofilaricidal treatment (the 'test' group) or ivermectin (the 'control' group, Fig. 1). The design most closely mirrors a phase IIb trial that is focused on comparing the efficacy of a new drug (a macrofilaricide) with an existing one (ivermectin) but is more limited in size than a typical phase III trial (see for examples the phase II and III trials of moxidectin for the treatment of onchocerciasis)[50,52].

**Pharmacodynamics assumptions.** The hypothetical macrofilaricides were assumed to be either purely macrofilaricidal, with no effect on mf, or macrofilaricidal and microfilaricidal. These are referred to, respectively, as macrofilaricidal-only macrofilaricides (MOMs) or macrofilaricidal and microfilaricidal macrofilaricides (MAMMs). Macrofilaricides were assumed to kill a fraction of adult worms over three months that was varied between 60% and 90% (Table 1). The MAMMs were assumed to have microfilaricidal effects identical to the microfilaricidal dynamics (but excluding the temporary sterilising or embryostatic effect) of ivermectin, i.e., rapid depletion of mf by 98–99% after 1–2 months[7].

**Simulated response dynamics.** Parasitological (mf) responses to ivermectin and hypothetical macrofilaricides in individual participants were simulated for 36 months after treatment. We incorporated a statistical model for the sampling of mf by skin snipping, assuming an aggregated distribution of mf within the skin to capture the relatively poor sensitivity of skin snipping, particularly at low intensities of (adult worm) infection[54]. The infection status (positive or negative by skin snip, >0 mf/mg skin) or intensity (>4 or >8 mf/mg skin) that defined eligibility as measured by taking 2 or 4 skin snips was varied (Table 1).

We repeated 1,000 simulations, with each simulation targeting recruitment of two 50-participant cohorts (note that for an increasing minimum infection intensity inclusion criterion, the chance of recruiting 50 eligible participants decreases, see Supplementary Fig. 4), for each parameter combination indicated in Table 1 (72 parameter combinations; 72,000 simulations). For each simulation, we calculated the mean difference between outcome measures (i.e., mean number of mf/mg of skin per participant or mean percentage of participants positive for mf) in the (macrofilaricide-treated), test, and (ivermectin-treated), control groups at time $\tau$, $\hat{D}(\tau)$ (i.e., the mean response in the control group subtracted from the mean response in the test group, e.g., Fig. 4), and the standard deviation within each group, $\hat{\sigma}_T(\tau)$ and $\hat{\sigma}_C(\tau)$ ($T$ = test, $C$ = control).

**Sample size estimation.** From the estimates of $\hat{D}(\tau)$, $\hat{\sigma}_T(\tau)$ and $\hat{\sigma}_C(\tau)$ (one estimate per simulation) we calculated Welch's $t$-statistic and the expected 'true' value of each parameter, $D(\tau)$, $\sigma_T(\tau)$ and $\sigma_C(\tau)$ to approximate the non-centrality and degrees of freedom parameters for the non-central $t$-distribution (assuming unequal variances). We confirmed that these (non-centrality; degrees of freedom) parameters approximated adequately the simulated distributions of Welch's $t$-statistic (Supplementary Figs. 5–7) and proceeded to estimate sample sizes (to the nearest 25) required to achieve an 80% probability (power) of detecting a superior response (positive difference) in macrofilaricide-treated, test participants compared to ivermectin-treated controls, assuming a type I error (false rejection of the null hypothesis of no positive difference) of 5% and 10% drop out per year of recruited participants.

Our approach to estimating sample sizes assumes that the simulated target of 50 participants per cohort provide an adequate approximation to the 'true' average response and that these quantities (mean and inter-participant variance) are consistent across changing sample sizes. In essence, this assumption rests on the trial itself having limited immediate impact on community-wide transmission (and therefore rates of reinfection). Sample sizes are thus interpreted qualitatively in terms of the directional (increasing/decreasing) influence of changing design-relevant parameter values. Further details on the sample size estimation approach are given in Supplementary Methods.

**Reporting summary.** Further information on research design is available in the Nature Research Reporting Summary linked to this article.

## Data availability

No external data were used. The individual-level simulated data generated during this study are available from the corresponding author on reasonable request.

## Code availability

The model code is freely available at https://github.com/mrc-ide/EPIONCHO-IBM-CTS. Instructions for installing and running the model in R (version 3.6.2) are available in the README.html file which can be accessed at https://mrc-ide.github.io/EPIONCHO-IBM-CTS/README.html.

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

## Acknowledgements

M.W., J.I.D.H., and M.G.B. gratefully acknowledge funding from the NTD Modelling Consortium (https://www.ntdmodelling.org, grant number OPP1053230) by the Bill and Melinda Gates Foundation (https://www.gatesfoundation.org/) in partnership with the Task Force for Global Health (https://www.taskforce.org). M.W. and M.G.B. also acknowledge financial support from the Drugs for Neglected Diseases *initiative* (DND*i*, https://www.dndi.org). P.M. is supported by a UK Medical Research Council doctoral training award. M.G.B. acknowledges joint Centre funding from the UK Medical Research Council and Department for International Development (grant number MR/R015600/1).

## Author contributions

M.W., M.G.B., and B.P. conceived the analysis; M.W., J.I.D.H., and P.M. wrote the model code; M.W. wrote the draft manuscript; M.W., M.G.B., and B.P. wrote the final manuscript; M.W., F.M., B.P., and M.G.B. participated in discussions on the work; M.W., J.I.D.H., P.M., F.M., B.P., and M.G.B. read and approved the final manuscript.

## Competing interests

F.M. and B.P. work at the organisation (DND*i*) that sponsored this study. However, the study sponsor did not have any influence in the study design, analysis, interpretation of results, the writing of the manuscript, or the decision to submit the paper for publication, other than critical, independent and scientific contribution. No payment was received from any other funding sources, or pharmaceutical company. The corresponding author had final responsibility for decision to submit for publication. The remaining authors declare no competing interests.
