## [Peer Review File · Nature Communications]

Reviewers' Comments:

Reviewer #1:

Remarks to the Author:

This is a very interesting paper and I believe it has great potential. See the detailed report for some questions and comments on how it could be improved.

Report on designing antifilarial drug trials using clinical trial simulators: the case of river blindness

The major claims made by the paper are: 1) The use of Clinical Trial Simulators (CTS) have not been widely used to inform trial design for neglected tropical diseases while 2) they could be highly beneficial and 3) key design parameters (key in terms of their effects on power) for potential trials of a hypothetical macrofilaricidal drug are identified.

To the best of my knowledge the above claims are novel and will they be of interest to others in the community and the wider field. That being said, I believe there is some minor room for making the more work convincing (see below a list of suggestions to possibly strengthen the conclusions). Specially, claims 2) and 3). I feel that the paper, with some editions, has potential to influence thinking in the field.

1. The authors claim that the individual-based transmission model used is novel. How or why is this novel? This was not clear to me as an outsider to the specific NTD considered. What is the added value of its novelty? This could be discussed somewhere in the paper, even if briefly.
2. The second paragraph on page 4 is not entirely correct to me. I would not use the words simulation and modelling (or model-based) interchangeably. In clinical trials, statistical aspects of complex designs are very often assessed by simulations. Take for example Bayesian Adaptive Designs (insert reference). In that context, simulation is forward-looking while modelling is backward-looking, and while they share many common features these two are distinct concepts. I would re-write that paragraph to make this distinction.
3. The paper mentions a phase II-III two-arm clinical trial but I fail to see how this works. What is the question looked at in Phase II in this case (is it a dose-selection? It is selection of different compounds) and how does this feed in into the phase III design? More importantly, how does this impact power calculations. From reading the paper, it feels that the power reported is for the Phase III part only but I wonder what benefits from the simulations could impact the phase II part of the trial (as this can indirectly affect power of designs).
4. The word “optimal” and “optimising” is used too loosely as there are many ways to optimise a trial (power/efficiency being one of them). Alternative objectives for optimising a trial design can be considered (for example, a compromise between power and ethical characteristics, e.g. the number of patients best treated).
5. Why is prevalence fixed at 40% and why do the simulations do not assess the effect of changing this assumption? I presume this should impact design choices too.
6. Similarly, what is the participation rate assumed (those consenting to be randomised). 50 subjects out of how many agree to be randomised and is this sensible? 10% drop-out rate, how does this affect the power of the study and what happens if these two rates are different? These two parameters in practice are major causes of difficulties

for trials, so assessing their impact (for the feasibility of the trial) would also be highly valuable.

7. The authors say as one of their conclusions in the discussion: “The high variability in estimated sample sizes required to detect superiority [...] reflects the potential resource-saving benefits and efficacy gains that model-based trial design can offer.”

I see two caveats to this conclusion. First, is it model based or is it the model-based simulated trial design approach that would offer this advantage. I think it is the combination of a realistic transmission model coupled with the simulations of a trial design (and its features) that offer this benefit. So, it goes beyond the modelling.

Second, would this high variability also suggest that traditional fixed designs are too risky in this context as if we get the assumptions wrong a trial can go very easily from being feasible to not being so. After all, most of these are assumptions on which there will be little good evidence before the trial starts. Would it not be fair that these results suggest that an Adaptive approach to trial design (where assumptions may be assessed at interim points and decisions on the trial could be made) would potentially be the approach that could materialise such potential gains shown by this work?

The statistical analysis presented in the supplementary material is appropriate and valid. The only caveat to this is that I do not see how or if there is a Phase II stage in this simulation as the section describing the trial in the supplementary does not mention details of this at all.

Reviewer #2:

Remarks to the Author:

General comments:

The authors present a clinical trial simulator (CTS) for antiparasitic drugs, illustrate what it can do, and make a case for using the CTS to maximise the efficiency of trials. I am convinced of the value of the CTS and appreciate the Herculean effort involved in developing it and the succinctness of the description. Some important details were lost (or moved to supplements) and should be included in the main text to complete the story (details below). Other than that, I applaud the efforts of the authors, the product, and the way they described it. Finally, I have some suggestions for further discussion (at the end).

Introduction:

In the first paragraph, the authors explain "Ivermectin and DEC have potent activity against the microfilarial progeny (the stage transmitted to vectors) of adult worms (macrofilariae)", without mentioning the microfilaricidal effects against LF (which is briefly mentioned between brackets in the second paragraph).

I was a little confused by the order of information on progress towards elimination of oncho in the Americas and Africa (2nd paragraph: "good progress"), then LF globally (2nd paragraph: "even better than oncho"), and then again oncho in Africa (3rd paragraph: "still problematic"). It would be helpful for the reader if the bits on oncho are put together in one go.

The first sentence of the 4th paragraph lacks a period at the end.

Last part of last sentence of 5th paragraph should be "(and *can be* accounted for using trial simulators *that* explicitly model transmission)" instead of "(and *is* accounted for using trial simulators *which* explicitly model transmission)".

Results:

Table 1 provides a nice overview of the model parameters that were varied, but I find myself staring at it because of suboptimal formatting (a mass of bullet points on the right with a bunch of titles to the left of it that don't line up nicely with the bullets because of awkward line spacing). If allowed by the journal, it would help the eye to harmonise the line spacing and shade (on-and-off) sets of rows pertaining to the same parameter.

Further regarding Table 1: I think specialists in the field (and myself) would appreciate a little more detail about the assumed (model mechanics for the) pharmacodynamics? E.g. how is the microfilaricidal effect defined (life-span reduction, instantaneous removal)? What proportion of mf are killed? Are embryostatic effect included (i.e. a temporary stop in mf production by the female worm(s), how long)? Is variation in individual treatment responses considered (this only becomes clear in the discussion: "No consideration is given to inter-participant variation in treatment responses which would inflate sample sizes")? Later on in the results section, reference is made to "transient effects of ivermectin", so it would be good to clarify what they are in Table 1 (I couldn't find any details in the methods section of the main paper either). And please clarify whether the to what extent the microfilaricidal effects of the MAMM drug were assumed to be the same as for ivermectin (this is clarified in the methods section, but would we good to also have in Table 1 for clarity).

From Figure 2 and the text it is not clear to me what is assumed about the sexual reproduction of worms in the human host, which is especially relevant when assessing microfilaricidal drugs as treatment may result in non-reproductive single-sex infections. Are male worms explicitly modelled as well? The figure legend only mentions female worms, suggesting only females are modelled whereas

the methods section also mentions male worms. Or is the mating process represented in some other (deterministic) way and if so, does that process account for stochastic effect appropriately?

The following sentence seems to be missing some words (or contain too many): "A full methodological description is given in the Methods section and the mathematical details of EPIONCHO-IBM, including the adaptations implemented to yield a CTS, are given in Hamley et al. [46] and the Supplementary Information, EPIONCHO-IBM".

Figure 3: the grey lines/area in the plots that represents the infection levels in the community from which trial participants were selected is barely visible because of the patterns in the trial arms and the solid aspect of their visualisation. This could be improved by making the lines/areas in the plot semi-transparent. Further, please clarify in the figure legend how large the community was from which these 2x50 participants were selected (the relative difference with size of trial arms will determine the transmission effects of the trial in the wider community) and whether they were treated at all or not (e.g. as part of regular MDA; I assume not by the look of the grey pattern).

Figure 4: a very efficient representation of trial outcomes, but it's quite difficult for me to distinguish the two lightest of the three colours representing different levels of macrofilaricidal effects (. I suggest to use different colors, and/or space the bullets and bars for the three levels further apart, and/or add black lines around the shapes.

Sentence that is missing a closing bracket: "The CTS approach permits exploration of how the inclusion criterion applied for participation (either $x > 0$ mf/mg or $x > 2$ mf/mg..."

Legend to Figure 5: please explain the meaning of the colouring (green = better because lower sample size requirement) and indicate what sample sizes were actually simulated (probably a finite set to limit computation time). Also, it possible to further specify the sample size requirement " ≤ 50 "? Right now, it seems that there is no difference between taking 2 or 4 skin snip per person, because many of the best trial designs require " ≤ 50 " participants per arm. But if a design based on 2 snips per person required say 40 persons and a design based on 4 snips only requires 20 persons, that is information one would like to consider when designing the trial (e.g. when finding eligible participants is difficult). The statement that "Sample sizes decrease with increasing macrofilaricidal efficacy, and marginally (because of the increased measurement precision) for 4 versus 2 skin snips" is now entirely based on the simulation results for MOM and MAMM drugs with "low" or "medium" efficacy. For the highly efficacious drugs, it might actually matter more as the absolute variation in treatments effects is lower due the higher macrofilaricidal effect (and if that is not the case, the reader might actually still want to see that – this one at least). I therefore suggest to also simulate sample sizes 20, 30 and 40, in addition to the current lowest value of 50.

Discussion:

"The comparatively large required sample sizes and long follow-up times for the hypothetical MOM, represents..." should read "The comparatively large required sample sizes and long follow-up times for the hypothetical MOM [no comma] represent [no -s]..."

How do the estimated sample sizes reflect the reality of annual MDA campaigns that will probably continue during a trial duration of 12 to 24 months? Was this considered in the simulations and/or can it be accounted for?

In the statistical analysis framework within their CTS, why do the authors not consider an analysis comparing difference in differences based on individual dynamics of mf intensity over time? As I understand from the rest of the manuscript, the power estimates are solely based on comparing group

means or proportions at follow-time without taking into account of individuals pre-treatment mf densities (other than selecting participants with some minimal skin mf density). Might comparing the group-wise difference in average of individuals differences over time not provide more information? Or is there previous work that rules out this approach?

Based on the study results, can an argument be made for preferably testing MOM drugs in trials comparing MOM+ivermectin to ivermectin alone, i.e. to emulate the effect of a MAMM, which according to the results of this study has several advantages in terms of follow-up time and required trial arm size? I'm blissfully ignorant about whether that is even possible within the regulations for drug development. Still I think it is an interesting question that might be worth discussing.

Methods:

Sentence with excess word(s): "We assumed that the all individuals within the..."

What is the rationale for the eligibility criterion regarding the minimum number of mf/mg skin? Is the difference between >0 and >2 relevant? Is there some clinical or epidemiological consideration here that is not being explained? As mf densities are typically highly overdispersed between individuals, wouldn't selecting densities of e.g. >10 mf/mg skin be feasible as well, and might this not make a much more noticeable difference for the required sample size?

The authors explain that they simulate 50-person cohorts per trial arm, but from figure 5 it seems they simulated many more trial arm sizes.

Regarding the description of statistical test(s) used to determine whether a there was significant difference between groups in the simulation: did the authors in fact perform a T-test assuming unequal variances, or is this approach starting from summary statistics (difference between group means, and group-specific standard deviations) an approximation? And if so, why this approximate approach instead of the T-test itself? And were mf densities transformed (e.g. logarithmically) before group comparison to resolve non-normality of what I image to be highly skewed responses? Also, (why) is this approach adequate for comparing differences in mf prevalence between groups? Wouldn't a Fisher's exact test be more appropriate?

Luc E. Coffeng, MD PhD
Department of Public Health
Erasmus MC, University Medical Center Rotterdam
The Netherlands

Reviewer #1

Reviewer's comment 1.1. This is a very interesting paper and I believe it has great potential. See the detailed report for some questions and comments on how it could be improved.

Authors' response 1.1. We thank the Reviewer for the appreciative remarks about our work.

Reviewer's comment 1.2. The major claims made by the paper are: 1) The use of Clinical Trial Simulators (CTS) have not been widely used to inform trial design for neglected tropical diseases while 2) they could be highly beneficial and 3) key design parameters (key in terms of their effects on power) for potential trials of a hypothetical macrofilaricidal drug are identified. To the best of my knowledge the above claims are novel and will they be of interest to others in the community and the wider field. That being said, I believe there is some minor room for making the more work convincing (see below a list of suggestions to possibly strengthen the conclusions). Specially, claims 2) and 3). I feel that the paper, with some editions, has potential to influence thinking in the field.

Authors' response 1.2. We thank the Reviewer for the appreciative synopsis of the key findings and messages of our work.

Reviewer's comment 1.3. The authors claim that the individual-based transmission model used is novel. How or why is this novel? This was not clear to me as an outsider to the specific NTD considered. What is the added value of its novelty? This could be discussed somewhere in the paper, even if briefly.

Authors' response 1.3. We thank the Reviewer for this important comment. The clinical trial simulator (CTS) is adapted from our newly developed transmission dynamics model, EPIONCHO-IBM (Hamley et al. *PLoS Negl Trop Dis* 13(12):e0007557). Novel adaptations include: incorporation of the antifilarial action of hypothetical microfilaricides; simulation of the number of pregnant or breastfeeding women in an endemic community (who are excluded from the trial), and simulation of infection-level eligibility criteria so that only individuals with a minimum number of microfilariae are considered eligible. Moreover, the application is novel; this is the first time a transmission dynamics model has been used to simulate prospectively a clinical trial for a neglected tropical disease (NTD). The added value of using a transmission dynamics model for this purpose is that it captures explicitly the dynamics of reinfection (and repopulation of the skin by microfilariae) of parasites not exposed to drug treatment, an inevitability for trials conducted in disease-endemic settings. Without a transmission dynamics model, it is extremely difficult to predict outcome measures of drugs that elicit protracted pharmacodynamics—frequently the case for drugs used to treat filarial infections—because of the diluting effect on the drug response of reinfection by drug-naïve parasites. We have revised the manuscript to emphasise further the novelty of the CTS and its application:

“The CTS includes novel components required to simulate a clinical trial, including the (modifiable) antifilarial action of macrofilaricides and explicit simulation of factors defining participant eligibility.”

“The transmission model that underpins the CTS accounts for the inevitable reinfection of participants by drug-naïve parasites in endemic settings. Crucially, this permits *a priori* identification of opportune follow-up times after treatment when the balance between measurable drug effects and reinfection is most favourable. Moreover, the individual-based structure of the transmission model permits explicit simulation (and exploration) of other proposed protocols, such as the use of minimum infection level eligibility criteria and sampling procedures (e.g. parasitological sampling by skin snips). Explicit simulation of transmission and trial protocols represent the key benefits of using a CTS to inform trial design.”

Reviewer's comment 1.4. The second paragraph on page 4 is not entirely correct to me. I would not use the words simulation and modelling (or model-based) interchangeably. In clinical trials, statistical aspects of complex designs are very often assessed by simulations. Take for example Bayesian Adaptive Designs (insert reference). In that context, simulation is forward-looking while modelling is

backward-looking, and while they share many common features these two are distinct concepts. I would re-write that paragraph to make this distinction.

Authors' response 1.4. We thank the Reviewer for this comment. We agree that it is useful to clarify that by simulation, we mean forward-looking projection/prediction of trial outcomes. We have amended the paragraph such that it now reads:

“Clinical trial simulation is the (mathematical) representation of clinical trials to inform decision-making on trial design by forward projecting likely trial outcomes³⁹ (e.g. Vegvari et al.)⁴⁰ Since the early 2000s, simulation has been widely implemented in the pharmaceutical sector to assist with the design of trials, often searching for a balance between the cost of collecting data and the information that these data will provide on key aspects (such as safety and/or efficacy) of the drug under consideration.⁴¹ Clinical trial simulation has not been used to assist drug development in the NTD domain, although simulation studies have been used widely to project the impact of population-level interventions on onchocerciasis, LF and other NTDs,⁴² and to inform the design of cluster intervention trials targeting elimination of soil-transmitted helminthiases.^{43,44} Simulation offers the potential to inform the design of clinical trials with the subsequent resource savings particularly attractive to the development of drugs for NTDs.”

Reviewer's comment 1.5 The paper mentions a phase II-III two-arm clinical trial but I fail to see how this works. What is the question looked at in Phase II in this case (is it a dose-selection? It is selection of different compounds) and how does this feed in into the phase III design? More importantly, how does this impact power calculations. From reading the paper, it feels that the power reported is for the Phase III part only but I wonder what benefits from the simulations could impact the phase II part of the trial (as this can indirectly affect power of designs).

Authors' response 1.5. We thank the Reviewer for this important point. As the Reviewer highlights, phase II trials are typically designed to find the optimal dose of a new treatment (that has been found to be safe and acceptably tolerated in previous, typically phase I trials), while phase II and III trials are larger and focused on comparing the efficacy of a new treatment to an existing one. We had used the phase II/III term to indicate that the focus of the simulation is on identifying optimal designs to demonstrate superior efficacy of a novel macrofilaricide compared to ivermectin (i.e. a focus on efficacy with a comparator group, akin to a phase II/III trial) while also considering a trial conducted in a single setting using a limited (but statistically robust) number of participants (i.e. more similar to a phase II trial). However, we also note that it is commonplace for phase II trials to also include a comparator group treated with an existing therapy (see for examples in filarial diseases Awadzi et al. *PLoS Negl Trop Dis* 2014 26, e2953 <https://www.clinicaltrials.gov/ct2/show/NCT00300768>; Thomsen et al. *Clin Infect Dis* 2016, 62, 334-341 <https://clinicaltrials.gov/ct2/show/NCT01975441>; King et al. *N Engl J Med* 2018, 8, 1801-1810 <https://clinicaltrials.gov/ct2/show/NCT01975441>). Our simulations most accurately reflect a phase IIb trial, which is restricted in participant numbers but has a focus on efficacy assessment. We have amended the manuscript to reflect this. The relevant sections of the revised manuscript now read:

“We use an adaptation of our new individual-based onchocerciasis transmission model, EPIONCHO-IBM,⁴⁵ to simulate a hypothetical phase IIb two-arm clinical trial (i.e. a trial focused on assessing efficacy but more limited in size than a phase III trial) comparing the efficacy of the hypothetical macrofilaricide to ivermectin.”

“The design most closely mirrors a phase IIb trial that is focused on comparing the efficacy of a new drug (a macrofilaricide) with an existing one (ivermectin) but is more limited in size than a typical phase III trial (see for examples the phase II and III trials of moxidectin for the treatment of onchocerciasis).^{46,48}”

“We have illustrated how a CTS can be used to inform the design of antifilarial drugs, using as an example a simple two-arm (phase IIb) trial of a hypothetical macrofilaricide compared to ivermectin for the treatment of onchocerciasis.”

Reviewer’s comment 1.6. The word “optimal” and “optimising” is used too loosely as there are many ways to optimise a trial (power/efficiency being one of them). Alternative objectives for optimising a trial design can be considered (for example, a compromise between power and ethical characteristics, e.g. the number of patients best treated).

Authors’ response 1.6. We have tightened our use of the words “optimal” and “optimising” and removed any loose descriptions. Relevant passages of the revised manuscript now read:

“Clinical trial simulators (CTSs) project patient outcomes to inform the design of clinical trials but have not been widely applied to NTDs, where their resource-saving payoffs could be highly beneficial.”

“Since the early 2000s, simulation has been widely implemented in the pharmaceutical sector to assist with the design of trials, often searching for a balance between the cost of collecting data and the information that these data will provide on key aspects (such as safety and/or efficacy) of the drug under consideration.⁴¹”

“Simulation offers the potential to inform the design of clinical trials with the subsequent resource savings particularly attractive to the development of drugs for NTDs.”

“...we discuss our results in the context of the definition and refinement of TPPs, the need for improved diagnostic indicators (biomarkers) of patent onchocerciasis, and the implementation of trials within a landscape of MDA.”

Reviewer’s comment 1.7. Why is prevalence fixed at 40% and why do the simulations do not assess the effect of changing this assumption? I presume this should impact design choices too.

Authors’ response 1.7. We had originally fixed the baseline microfilarial prevalence to 40% to reflect that the majority of ivermectin-naïve communities with greater than 40% prevalence have been under mass drug administration for many (but variable) years. We explain in the Discussion of the revised manuscript that finding ivermectin-naïve communities for the purposes of clinical trials is desirable because of the uncertain effects of multiple prior rounds of ivermectin on *Onchocerca volvulus* in those situations in which ivermectin MDA has been implemented and the difficulty with measuring accurately how many prior treatments individuals have received. (Although, with caution, the simulator can be used to predict response dynamics in communities with a prior history of MDA; this is illustrated in the README.html file that accompanies the simulator code, see also **Authors’ 2.17** for further details.) Notwithstanding these considerations, and as suggested by the Reviewer, we have also included in the revised manuscript, simulations conducted with a baseline microfilarial prevalence of 50%. Realistically, it is unlikely that there exist many treatment-naïve transmission settings with greater than 50% microfilarial prevalence (because priority ivermectin MDA has focused on meso- and hyperendemic communities). Relevant sections of the revised manuscript—including reference to new figures and text included in the Supplementary Information—now read:

“We focus on trials conducted in previously ivermectin-naïve, mesoendemic transmission foci with a microfilarial prevalence among individuals aged ≥ 5 years ranging from 40% to 50% (transmission foci with higher microfilarial prevalence are likely to have been undergoing MDA for many years).”

“We developed a CTS from the individual-based onchocerciasis transmission model EPIONCHO-IBM⁴⁵ to simulate a trial conducted in an ivermectin-naïve mesoendemic community with a microfilarial prevalence in those aged ≥ 5 years of either 40% or 50%.”

“There is negligible qualitative difference in the treatment response dynamics between the lower and upper end of the mesoendemicity setting (compare **Figure 3** with **Figure S1** in the Supplementary Information).”

“Inference on opportune follow-up timeframes is qualitatively very similar between the lower and the upper end of the mesoendemicity setting, albeit there is

noticeably greater variation in the difference between the microfilarial intensity outcome measure at the upper endemicity end of the setting (compare **Figure S2a** and **S2b** with **Figure 4a** and **4b**).”

“By contrast, for microfilarial prevalence, increasing the infection level inclusion criterion increases the sample size. This is because the selection of more heavily infected participants decreases the number of participants who are ostensibly ‘cured’ (i.e. achieving zero mf) following treatment (note that macrofilaricidal efficacy is defined probabilistically as the chance that an adult *Onchocerca volvulus* worm is killed by treatment). These contrasting effects are enhanced in the upper-end mesoendemic setting because participants tend to be more heavily infected (Supplementary Information, Figure S3).”

Reviewer’s comment 1.8. Similarly, what is the participation rate assumed (those consenting to be randomised). 50 subjects out of how many agree to be randomised and is this sensible? 10% drop-out rate, how does this affect the power of the study and what happens if these two rates are different? These two parameters in practice are major causes of difficulties for trials, so assessing their impact (for the feasibility of the trial) would also be highly valuable.

Authors’ response 1.8. We do not model explicitly the proportion of potentially eligible individuals who consent to be screened for inclusion in the trial, although we highlight in the revised manuscript that this could be implemented and would be useful for situations where participation fatigue may be important. This could be the case in communities where awareness of the disease has diminished following years of mass drug administration that has effectively controlled onchocerciasis as a public health problem. Similarly, we do not vary explicitly the 10% per year drop-out rate, albeit emphasising in the revised manuscript that this is a model parameter that can be changed as required. We highlight how this is done in the README.html file that accompanies the clinical trial simulator code (which we had not mentioned in the original version of the manuscript). The relevant section of the revised manuscript now reads:

“Moreover, in communities where awareness of onchocerciasis has diminished after years of MDA that has effectively controlled the disease as a public health problem, the challenges of recruiting and retaining participants during the trial may increase. It would be straightforward to adapt the CTS to model the proportion of individuals who consent to be screened for infection (by skin snipping). The dropout rate (which was here set to a nominal 10% per year) is already included as a parameter that can be varied (see the README.html file within the code repository at <https://github.com/mrc-ide/EPIONCHO-IBM-CTS>).”

The use of 50-participants per treatment cohort (from a total population of 1,000 individuals) is used for the purposes of (a) illustrating the response dynamics following treatment with a macrofilaricide or ivermectin (see Figure 3), and (b) estimating the mean and variance (standard deviation) in responses to enable approximation of the required sample sizes. This approach avoided having to simulate explicitly numerous cohort sizes to calculate sample sizes numerically which would have been computationally prohibitive (see also **Authors’ response 2.2** for further details). We clarify this approach in the Methods section of the revised manuscript, with complete details given in the Supplementary Information. Relevant sections of the revised Methods section now read:

“We assumed that everyone aged ≥ 18 years in a population of 1,000 individuals who were not pregnant or breastfeeding were screened for mf (by skin snipping). We further assumed that a set (target) number of eligible individuals consented to participate (we targeted 50 participants per cohort for the purpose of approximating the mean and variance among individual response dynamics to estimate sample sizes).”

“We repeated 1,000 simulations, with each simulation targeting recruitment of two 50-participant cohorts (note that for an increasing minimum infection intensity inclusion criterion, the chance of recruiting 50 eligible participants decreases, see Supplementary Information Figure S4), for each parameter combination indicated in **Table 1** (72 parameter combinations; 72,000 simulations). For each simulation, we calculated the mean difference between outcome measures (i.e. mean number

of mf/mg of skin per participant or mean percentage of participants positive for mf) in the (macrofilaricide-treated), test, and (ivermectin-treated), control, groups at time τ , $\widehat{D}(\tau)$ (i.e. the mean response in the control group subtracted from the mean response in the test group, e.g. **Figure 4**), and the standard deviation within each group, $\widehat{\sigma}_T(\tau)$ and $\widehat{\sigma}_C(\tau)$ (T = test, C = control).”

“From the estimates of $\widehat{D}(\tau)$, $\widehat{\sigma}_T(\tau)$ and $\widehat{\sigma}_C(\tau)$ (one estimate per simulation) we calculated Welch’s t -statistic and the expected ‘true’ value of each parameter, $D(\tau)$, $\sigma_T(\tau)$ and $\sigma_C(\tau)$ to approximate the non-centrality and degrees of freedom parameters for the non-central t -distribution (assuming unequal variances). We confirmed that these (non-centrality; degrees of freedom) parameters approximated adequately the simulated distributions of Welch’s t -statistic (Supplementary Information, Figure S5–S7) and proceeded to estimate sample sizes (to the nearest 25) required to achieve an 80% probability (power) of detecting a superior response (positive difference) in macrofilaricide-treated, test, participants compared to ivermectin-treated controls, assuming a type I error (false rejection of the null hypothesis of no positive difference) of 5% and 10% drop out per year of recruited participants.”

Reviewer’s comment 1.9. The authors say as one of their conclusions in the discussion: “The high variability in estimated sample sizes required to detect superiority [...] reflects the potential resource-saving benefits and efficacy gains that model-based trial design can offer.” I see two caveats to these conclusions. First, is it model based or it is the model-based simulated trial design approach that would offer these advantages. I think it is the combination of a realistic transmission model coupled with the simulations of a trial design (and its features) that offer this benefit. So, it goes beyond the modelling. Second, would this high variability also suggest that traditional fixed designs are too risky in this context as if we get the assumptions wrong a trial can go very easily from being feasible to not being so. After all, most of these are assumptions on which there will be little good evidence before the trial start. Would it not be fair that these results suggest that an Adaptive approach to trial design (where assumptions may be assessed at interim points and decisions on the trial could be made) would potentially be the approach that could materialise such potential gains shown by this work?

Authors’ response 1.9 We wholly agree with the Reviewer that there are two fundamental strengths of using a simulator to inform the design of a trial. First, that the underlying transmission dynamics model can capture explicitly the effects of reinfection in disease-endemic communities. Second, that the simulator (here an individual-based transmission model) can capture various aspects of the trial design as parameters that can be varied to identify favourable choices (e.g. appropriate follow-up times, infection eligibility criteria and parasitological sampling protocols). We have emphasised these key benefits in the revised version of the manuscript:

“The transmission model that underpins the CTS accounts for the inevitable reinfection of participants by drug-naïve parasites in endemic settings. Crucially, this permits *a priori* identification of opportune follow-up times after treatment when the balance between measurable drug effects and reinfection is most favourable. Moreover, the individual-based structure of the transmission model permits explicit simulation (and exploration) of other proposed protocols, such as the use of minimum infection level eligibility criteria and sampling procedures (e.g. parasitological sampling by skin snips). Explicit simulation of transmission and trial protocols represent the key benefits of using a CTS to inform trial design.”

“The variability in estimated sample sizes required to detect superiority of a macrofilaricide compared to ivermectin reflects the potential resource-saving benefits and efficacy gains that clinical trial simulation can offer.”

We also completely agree with the Reviewer that our results suggest that adaptive approaches to trial designs could be extremely beneficial, especially for mid-trial refinements to the trial design and for making early ‘go/no go’ decisions. For example, sampling participants at interim time points—*before* the initially-proposed most opportune follow-up time—would permit the clinical trial simulator to be fitted to the data to achieve an early indication/estimate of macrofilaricidal activity and accompanying microfilaricidal activity. This information could be used to: (a) decide whether to continue or not with the trial (e.g. if insufficient macrofilaricidal activity is detected) and (b) refine the choice of follow-up

time for the principal outcome measure. We had alluded to the use of adaptive approaches in the original version of the manuscript but we have strengthened and make more explicit our discussion of this important issue in the revised version:

“An alternative approach to earlier identification of macrofilaricidal activity is to collect interim parasitological (microfilarial) data before the indicated optimal follow-up time. While these data may not indicate superiority compared to ivermectin, model-based interpretation of the data (for example by fitting the CTS or related transmission dynamics models)^{8,55} will give an indication of the underlying activity and efficacy of the trialled drug. This may give important early indications to support ‘go/no go’ decisions and/or inform refinement or adaption of the trial design. For example, measuring mf from patients at interim time points—*before* the initially-proposed most opportune follow-up time—would permit the CTS to be fitted to the data to achieve an early indication/estimate of macrofilaricidal activity and accompanying microfilaricidal activity. Decisions could then be made on whether to continue the trial (e.g. depending on how the early estimate of macrofilaricidal activity compares to the desired TPP) and when the final follow up should be made (e.g. depending on the indicated level of microfilaricidal activity). The efficiency gains associated with such adaptive (as opposed to ‘fixed’) trial designs is well recognised and has led to their increased use across the clinical medicine domain.^{71,72} Collection of interim longitudinal repeated measurements also confers added statistical power to post hoc analyses and can supplement the final comparison of trialled treatments.”

Reviewer’s comment 1.10. The statistical analysis presented in the supplementary material is appropriate and valid. The only caveat to this is that I do not see how or if there is a Phase II stage in this simulations as the section describing the trial in the supplementary does not mention details of this at all.

Authors’ response 1.10. We thank the Reviewer for the supportive appraisal of our statistical approach. We have clarified throughout the manuscript that our simulations most closely mimic a phase IIb clinical trial, which focuses on assessing efficacy but is more limited in size than a typical phase III study (see **Authors’ response 1.5**). All relevant information on the simulated study design is now contained within the main text of the revised manuscript and we limit the Supplementary Information to technical detail of the model and the statistical approach. In particular, we have bolstered our description of the trial design in the Methods section of the revised manuscript:

“We randomly assigned consenting participants to receiving either a hypothetical macrofilaricidal treatment (the ‘test’ group) or ivermectin (the ‘control’ group, **Figure 1**). The design most closely mirrors a phase IIb trial that is focused on comparing the efficacy of a new drug (a macrofilaricide) with an existing one (ivermectin) but is more limited in size than a typical phase III trial (see for examples the phase II and III trials of moxidectin for the treatment of onchocerciasis).^{46,48}”

Reviewer #2

Reviewer’s comment 2.1. The authors present a clinical trial simulator (CTS) for antifilarial drugs, illustrate what it can do, and make a case for using the CTS to maximise the efficiency of trials. I am convinced of the value of the CTS and appreciate the Herculean effort involved in developing it and the succinctness of the description. Some important details were lost (or moved to supplements) and should be included in the main text to complete the story (details below). Other than that, I applaud the efforts of the authors, the product, and the way they described it. Finally, I have some suggestions for further discussion (at the end).

Authors’ response 2.1. We thank the Reviewer for this supportive appraisal of our work.

Reviewer’s comment 2.2. In the first paragraph, the authors explain “Ivermectin and DEC have potent activity against the microfilarial progeny (the stage transmitted to vectors) of adult worms

(macrofilariae)", without mentioning the macrofilaricidal effects against LF (which is briefly mentioned between brackets in the second paragraph).

Authors' response 2.2. We thank the Reviewer for raising this ambiguity. We have clarified this paragraph emphasising that combinations of diethylcarbamazine (DEC), albendazole with or without ivermectin have significant macrofilaricidal activity against adult filariae (as determined by levels of circulating filarial antigen).

"For LF, combinations of DEC and albendazole,⁹ and the new triple combination therapy, IDA (ivermectin with DEC and albendazole),⁴ has significant activity against adult worms."

Reviewer's comment 2.3. I was a little confused by the order of information on progress towards elimination of oncho in the Americas and Africa (2nd paragraph: "good progress"), then LF globally (2nd paragraph: "even better than oncho"), and then again oncho in Africa (3rd paragraph: "still problematic"). It would be helpful for the reader if the bits on oncho are put together in one go.

Authors' response 2.3. We thank the Reviewer for this suggestion. We have reorganised the material in paragraphs 2 and 3 which now reads:

"The suppression of mf elicited by antifilarial drugs—combined with their efficacy in single oral doses—makes them perfectly suited for lowering and potentially interrupting transmission of filarial infections by MDA if delivered at a high coverage and treatment adherence, and on either an annual or semi-annual basis.² For LF, the microfilaricidal (and macrofilaricidal) efficacy of the drugs, and the (assumed) shorter life-expectancy of the adult worm compared to *O. volvulus* (about 5 years compared to about 10 years), has led to the expectation that widespread elimination is feasible, particularly if MDA is combined with vector control.^{10,11} As of 2017, 21 out of 73 LF-endemic countries had stopped MDA and transitioned to post-treatment surveillance.⁶ For onchocerciasis, MDA has greatly reduced morbidity and excess mortality^{12,13} and has successfully eliminated onchocerciasis transmission from Colombia,¹⁴ Ecuador,¹⁵ northern Venezuela,¹⁶ Mexico,¹⁷ and Guatemala in Latin America.¹⁸ Good progress towards elimination has also been made in Africa,¹⁹ which bears 99% of the onchocerciasis cases, with notable successes in foci in Mali, Senegal^{20,21} Nigeria¹⁹ and Sudan.²²

However, the long-lived nature of adult *O. volvulus*, and the persistence of focal areas of intense transmission mean that elimination of onchocerciasis at a country scale will be extremely challenging within currently proposed timeframes.²³ Despite many years of MDA, transmission continues in highly endemic onchocerciasis communities with high vector biting rates.²⁴⁻²⁷

Reviewer's comment 2.4. The first sentence of the 4th paragraph lacks a period at the end.

Authors' response 2.4. We thank the Reviewer for spotting this. This is amended in the revised manuscript.

Reviewer's comment 2.5. Last part of last sentence of 5th paragraph should be "(and *can be* accounted for using trial simulators *that* explicitly model transmission)" instead of "(and *is* accounted for using trial simulators *which* explicitly model transmission)".

Authors' response 2.5. We have corrected the relevant sentence.

Reviewer's comment 2.6. Table 1 provides a nice overview of the model parameters that were varied, but I find myself staring at it because of suboptimal formatting (a mass of bullet points on the right with a bunch of titles to the left of it that don't line up nicely with the bullets because of awkward line spacing). If allowed by the journal, it would help the eye to harmonise the line spacing and shade (on-and-off) sets of rows pertaining to the same parameter.

Authors' response 2.6. We have removed the bullet points from the table to enhance the clarity. We will follow the journal's formatting rules should the manuscript be accepted for publication.

Reviewer's comment 2.7. Further regarding Table 1: I think specialists in the field (and myself) would appreciate a little more detail about the assumed (model mechanics for the) pharmacodynamics? E.g. how is the microfilaricidal effect defined (life-span reduction, instantaneous removal)? What proportion of mf are killed? Are embryostatic effect included (i.e. a temporary stop in mf production by the female worm(s), how long)? Is variation in individual treatment responses considered (this only becomes clear in the discussion: "No consideration is given to inter-participant variation in treatment responses which would inflate sample sizes")? Later on in the results section, reference is made to "transient effects of ivermectin", so it would be good to clarify what they are in Table 1 (I couldn't find any details in the methods section of the main paper either). And please clarify whether the/to what extent the microfilaricidal effects of the MAMM drug were assumed to be the same as for ivermectin (this is clarified in the methods section, but would we good to also have in Table 1 for clarity).

Authors' response 2.7. We thank the Reviewer for these suggestions. We have clarified in the opening paragraph of the Results how the pharmacodynamics of the macrofilaricides and ivermectin are modelled and included additional footnotes for Table 1. The opening Results paragraph now reads:

"Macrofilaricidal effects are modelled as the percentage of adult *O. volvulus* killed within three months of treatment. The microfilaricidal effect of a MAMM is assumed to be identical to that of ivermectin (but *without* the temporary sterilisation or so-called embryostatic effect, see Basáñez et al.⁷). Mathematical details of EPIONCHO-IBM, including a complete description of the PD properties of ivermectin, macrofilaricides and other adaptations implemented to yield the CTS, are given in the Supplementary Information, *Supplementary Methods*."

The footnotes for Table 1 now read:

^a Macrofilaricidal efficacy is defined as the probability (expressed as a percentage) that an adult *Onchocerca volvulus* worm is killed within three months of (single dose) treatment with the hypothetical drug.

^b Microfilaricidal effect is assumed to be identical to that of ivermectin (excluding the temporary sterilisation or so-called embryostatic effect, see Supplementary Information, *Supplementary Methods*).

In these revised passages, in the interests of a word limit, we refer the reader to the Supplementary Information for complete details on the mechanics of how the effects of the macrofilaricides and ivermectin are modelled. We also refer the reader explicitly to the paper by Basáñez et al. (*Lancet Infect Dis* 2008, 8, 310-322) where the model structure describing the effects of ivermectin on *Onchocerca volvulus* was first developed. The Reviewer correctly identifies that, as stated in the Discussion, no inter-participant variation in responses to either ivermectin or macrofilaricides is considered.

"No consideration is given to inter-participant variation in treatment responses⁶⁹ (PD variation) which would inflate sample sizes."

Reviewer's comment 2.8. From Figure 2 and the text it is not clear to me what is assumed about the sexual reproduction of worms in the human host, which is especially relevant when assessing microfilaricidal drugs as treatment may result in non-reproductive single-sex infections. Are male worms explicitly modelled as well? The figure legend only mentions female worms, suggesting only females are modelled whereas the methods section also mentions male worms. Or is the mating process represented in some other (deterministic) way and if so, does that process account for stochastic effect appropriately?

Authors' response 2.8. We thank the Reviewer for this important comment. The number of male worms per host is modelled explicitly and it is assumed that if one male worm is present in a host then

all females will produce microfilariae (i.e. a completely polygamous mating system). We have clarified this in the legend of Figure 2. The relevant sections now read:

“EPIONCHO-IBM is an individual-based analogue of the population-based (deterministic) EPIONCHO transmission model^{75,76} tracking the number and age (a) of adult non-fertile ($W_{FN(i,a)}$) and fertile ($W_{FF(i,a)}$) female *Onchocerca volvulus* worms, microfilariae ($M_f(a)$) and male worms (not represented schematically) within individual (human) hosts ...”

“Female worms in a host are assumed to produce microfilariae if at least one male worm is present (i.e. polygamous mating).”

Reviewer’s comment 2.9. The following sentence seems to be missing some words (or contain too many): “A full methodological description is given in the Methods section and the mathematical details of EPIONCHO-IBM, including the adaptations implemented to yield a CTS, are given in Hamley et al. [46] and the Supplementary Information, EPIONCHO-IBM”.

Authors’ response 2.9. We have clarified this sentence which now reads:

“Mathematical details of EPIONCHO-IBM, including a complete description of the PD properties of ivermectin, macrofilaricides and other adaptations implemented to yield the CTS, are given in the Supplementary Information, *Supplementary Methods*.”

Reviewer’s comment 2.10. Figure 3: the grey lines/area in the plots that represents the infection levels in the community from which trial participants were selected is barely visible because of the patterns in the trial arms and the solid aspect of their visualisation. This could be improved by making the lines/areas in the plot semi-transparent. Further, please clarify in the figure legend how large the community was from which these 2x50 participants were selected (the relative difference with size of trial arms will determine the transmission effects of the trial in the wider community) and whether they were treated at all or not (e.g. as part of regular MDA; I assume not by the look of the grey pattern).

Authors’ response 2.10. We thank the Reviewer for the suggestions to improve the clarity of Figure 2. We have ‘thinned’ the simulations, presenting the results at approximately half-monthly intervals. This makes the infection dynamics in the community more visible. We have also darkened the lines of the community dynamics, again improving clarity. We cannot use transparency as this is not supported by vector format graphics, the preferred format for the journal. We have clarified in the main text and in the legend of Figure 3 that the simulated population comprises 1,000 individuals and that (eligible) trial participants were treated (i.e. as the Reviewer correctly identifies, no mass drug administration was simulated). The relevant revised sections now read:

“We assumed that everyone aged ≥ 18 years in a population of 1,000 individuals who were not pregnant or breastfeeding were screened for mf (by skin snipping).”

“The grey lines indicate the mean (either mf per mg of skin, i.e. intensity, or presence of mf, i.e. prevalence) in the whole population, which is assumed to comprise 1,000 individuals. Note that only eligible participants are treated, i.e. (community-wide) mass drug administration is not simulated.”

Reviewer’s comment 2.11. Figure 4: a very efficient representation of trial outcomes, but it’s quite difficult for me to distinguish the two lightest of the three colours representing different levels of macrofilaricidal effects (. I suggest to use different colors, and/or space the bullets and bars for the three levels further apart, and/or add black lines around the shapes.

Authors’ response 2.11. We thank the Reviewer for the suggestions on how to improve the clarity of Figure 4. We have increased the spacing between the points and lines depicting the results from macrofilaricides with different efficacies. We have also increased the contrast between the colours.

Reviewer’s comment 2.12. Sentence that is missing a closing bracket: “The CTS approach permits exploration of how the inclusion criterion applied for participation (either $x > 0$ mf/mg or $x > 2$ mf/mg...”

Authors' response 2.12. This has been corrected.

Reviewer's comment 2.13. Legend to Figure 5: please explain the meaning of the colouring (green = better because lower sample size requirement) and indicate what sample sizes were actually simulated (probably a finite set to limit computation time). Also, it possible to further specify the sample size requirement " ≤ 50 "? Right now, it seems that there is no difference between taking 2 or 4 skin snips per person, because many of the best trial designs require " ≤ 50 " participants per arm. But if a design based on 2 snips per person required say 40 persons and a design based on 4 snips only requires 20 persons, that is information one would like to consider when designing the trial (e.g. when finding eligible participants is difficult). The statement that "Sample sizes decrease with increasing macrofilaricidal efficacy, and marginally (because of the increased measurement precision) for 4 versus 2 skin snips" is now entirely based on the simulation results for MOM and MAMM drugs with "low" or "medium" efficacy. For the highly efficacious drugs, it might actually matter more as the absolute variation in treatments effects is lower due the higher macrofilaricidal effect (and if that is not the case, the reader might actually still want to see that – this one at least). I therefore suggest to also simulate sample sizes 20, 30 and 40, in addition to the current lowest value of 50.

Authors' response 2.15. We thank the Reviewer for this comment. We have modified Figure 5 to include a legend which indicates how the colours relate to the estimated sample size, from green (smallest) to red (largest). We have clarified in the Methods how sample sizes were estimated (see also **Authors' response 1.8** and **Authors' response 2.22**). In particular, we used a method to approximate sample sizes by estimating the mean and variance (standard deviation) in responses from macrofilaricide-treated and ivermectin-treated controls in two 50-participant cohorts. Indeed, we targeted cohorts of 50 because our approximation method relies on our simulations of Welch's t -statistic being adequately described by a non-central t -distribution (which we confirm in the Supplementary Information; see also **Authors' response 2.22** and **Authors' response 2.23**). This approximation approach avoided having to simulate explicitly numerous cohort sizes to calculate sample sizes numerically which would have been computationally prohibitive. Because of this approximation approach, the sample sizes are best interpreted in a more qualitative manner, showing the direction of change as particular parameters are modified. For example, Figure 5, illustrates that there is a modest decline in required sample sizes under certain circumstances but that this is a relatively minor effect compared to changes associated with the choice of follow-up time or the main response outcome (microfilarial prevalence or intensity). We have reflected these arguments in the revised manuscript:

"Sample sizes decrease with increasing macrofilaricidal efficacy, and marginally for microfilarial intensity (because of the increased measurement precision) for 4 versus 2 skin snips. More pronounced effects result from the choice of outcome measure and infection level inclusion criteria. Sample sizes are generally lower when using microfilarial prevalence compared to microfilarial intensity for an inclusion criterion of > 0 mf/mg because of the lower inter-participant variability associated with the former. However, for an increasing minimum microfilarial intensity inclusion criterion (ranging from > 0 mf/mg to > 8 mf/mg) opposing directional effects on sample sizes for the different outcome measures are evident. For microfilarial intensity, increasing the infection level inclusion criterion decreases required sample sizes because inter-participant variability in microfilarial counts is reduced (by selecting individuals with somewhat more similar microfilarial loads). By contrast, for microfilarial prevalence, increasing the infection level inclusion criterion increases the sample size. This is because the selection of more heavily infected participants decreases the number of participants who are ostensibly 'cured' (i.e. achieving zero mf) following treatment (note that macrofilaricidal efficacy is defined probabilistically as the chance that an adult *Onchocerca volvulus* worm is killed by treatment). These contrasting effects are enhanced in the upper-end mesoendemic setting because participants tend to be more heavily infected (Supplementary Information, Figure S3)."

"Sample sizes are thus interpreted qualitatively in terms of the directional (increasing/decreasing) influence of changing design-relevant parameter values. Further details on the sample size estimation approach are given in Supplementary Information, *Supplementary Methods*."

Reviewer’s comment 2.16. “The comparatively large required sample sizes and long follow-up times for the hypothetical MOM, represents...” should read “The comparatively large required sample sizes and long follow-up times for the hypothetical MOM [no comma] represent [no -s]...”.

Authors’ response 2.16. This has been amended.

Reviewer’s comment 2.17. How do the estimated sample sizes reflect the reality of annual MDA campaigns that will probably continue during a trial duration of 12 to 24 months? Was this considered in the simulations and/or can it be accounted for?

Authors’ response 2.17. We thank the Reviewer for this important comment. The main objective of the manuscript is to illustrate how a clinical trial simulator (CTS) can be used to inform the design of trials of new (macrofilaricidal) antifilarial drugs. To achieve this, we have deliberately simulated a trial conducted in a treatment-naïve setting (but see also **Authors’ response 1.7**) without ongoing mass drug administration (MDA). This serves to convey the key messages on the utility and resource-saving payoffs of using the CTS (see **Authors’ response 1.3**). In reality, of course, mass drug administration would very likely be initiated in such a community on completion of the trial (if it had not already been started). It is perfectly feasible to simulate a trial conducted with ongoing mass drug administration. Indeed, we provide examples of this in the README.html file that accompanies the simulator code (which is freely available at <https://github.com/mrc-ide/EPIONCHO-IBM-CTS>). We discuss simulating clinical trials in a backdrop of MDA with explicit reference to these examples in the revised manuscript:

“While simulating a trial to be conducted in a backdrop of prior and ongoing rounds of MDA adds complexity and additional uncertainty, it is included as an additional feature of the CTS presented here (for examples see the README.html file). However, it is inevitable that the accuracy of the predictions will be diminished because of the additional assumptions required. For example, even given at annual MDA frequencies, ivermectin exerts notable (but uncertain) macrofilaricidal effects on *O. volvulus* and cumulative reductions in female worm fertility⁸ which will propagate into the projected response dynamics following treatment. Moreover, obtaining accurate and reliable information on how many prior rounds of treatment individual participants have received is a challenge, although information on how many rounds of MDA have been distributed at a population level is more readily available.”

Reviewer’s comment 2.18. In the statistical analysis framework within their CTS, why do the authors not consider an analysis comparing difference in differences based on individual dynamics of mf intensity over time? As I understand from the rest of the manuscript, the power estimates are solely based on comparing group means or proportions at follow-time without taking into account of individuals pre-treatment mf densities (other than selecting participants with some minimal skin mf density). Might comparing the group-wise difference in average of individuals differences over time not provide more information? Or is there previous work that rules out this approach?

Authors’ response 2.18. We agree with the Reviewer’s opinion that an individual-based longitudinal analysis is likely to provide a more powerful analytical approach than a viz-a-viz comparison of average responses between treatment arms. Indeed, such data, collected at multiple times after treatment would also be highly useful for modelling and may provide an early indication of the efficacy of a new macrofilaricide (see also our discussion of adaptive trial designs, **Authors’ response 1.9**) We address both of these points in the manuscript:

“An alternative approach to earlier identification of macrofilaricidal activity is to collect interim parasitological (microfilarial) data before the indicated optimal follow-up time. While these data may not indicate superiority compared to ivermectin, model-based interpretation of the data (for example by fitting the CTS or related transmission dynamics models)^{8,55} will give an indication of the underlying activity and efficacy of the trialed drug”

We have focused on illustrating how a clinical trial simulator can inform (more classical fixed) design decisions on efficacy assessment among groups of participants randomized to receive different treatments in a classical clinical trial structure. While more complex (and powerful) statistical/analytical approaches can supplement and reinforce final comparisons of trialled treatments, a clear and transparent demonstration of efficacy is a key requirement for regulators. We have raised this point in the revised version of the manuscript:

“The efficiency gains associated with such adaptive (as opposed to ‘fixed’) trial designs is well recognised and has led to their increased use across the clinical medicine domain.^{71,72} Collection of interim longitudinal repeated measurements also confers added statistical power to post hoc analyses and can supplement the final comparison of trialled treatments.”

Reviewer’s comment 2.19. Based on the study results, can an argument be made for preferably testing MOM drugs in trials comparing MOM+ivermectin to ivermectin alone, i.e. to emulate the effect of a MAMM, which according to the results of this study has several advantages in terms of follow-up time and required trial arm size? I’m blissfully ignorant about whether that is even possible within the regulations for drug development. Still I think it is an interesting question that might be worth discussing.

Authors’ response 2.19. We thank the Reviewer for this important question. We completely agree that a MOM combined with ivermectin would likely be an efficacious combination, combining macrofilaricidal properties with the microfilaricidal and embryostatic action of ivermectin (which would likely be more efficacious than a MAMM which we assume lacks the embryostatic effect of ivermectin, see **Authors’ response 2.7**). However, co-administration of a MOM with ivermectin would likely mean that the MOM be registered as a component of a combination therapy, which may not be entirely desirable (e.g. it may preclude the use of a MOM alone in onchocerciasis-loiasis co-endemic settings). We have discussed the possibility of trialling MOMs in conjunction with ivermectin (either as a combination therapy, or within ongoing MDA) in the revised manuscript:

“A seemingly natural solution to this problem is to design trials that concomitantly deliver the standard annual antifilarial therapy and compare outcomes in groups treated (not necessarily at the same time) with a macrofilaricide and a standard MDA treatment or treated with standard MDA alone. This type of strategy has been used in trials of anti-*Wolbachia* therapies for onchocerciasis and LF which elicit a similarly protracted decline in mf (caused by female worm sterilisation and eventual death following depletion of endosymbiotic *Wolbachia*), when used as monotherapy,⁴⁹ to the hypothetical MOM considered here, but which effect lasting suppression and lower rates of microfilarial repopulation after subsequent rounds of ivermectin treatment. Whether a MOM would be given at the same time as ivermectin, or with sufficient time before or after MDA rounds, would depend on the safety of co-administration and on whether or not it would be desirable to have a MOM registered exclusively as a component of a combination therapy with ivermectin.”

“Trials combining a MOM and ivermectin (whether or not as a combination therapy), however, may be challenging in areas co-endemic with the filarial parasite *Loa loa* because the killing of *L. loa* mf by microfilaricides in heavily microfilaraemic individuals can lead to severe and sometimes fatal severe adverse events (SAEs).²⁹ Indeed, the disruption to onchocerciasis and LF control caused by SAEs in communities co-endemic with loiasis is a key driver behind the current push to develop safe antifilarial drugs for loiasis-infected patients³² and novel test-and(not)-treat strategies.⁷⁰ In many such communities, individuals with high intensities of *L. loa* microfilaraemia must be identified and excluded from receiving microfilaricidal drugs (i.e. by test-and-not-treat strategies).⁷⁰ Hence, there remains a potential use case for MOM monotherapy (assuming a MOM does not affect *L. loa* mf) or indeed anti-*Wolbachia* therapies (that do not affect *L. loa* because they do not harbour *Wolbachia* endosymbionts) in onchocerciasis-loiasis co-endemic communities.”

Reviewer's comment 2.20. Sentence with excess word(s): "We assumed that the all individuals within the..."

Authors' response 2.20. We have revised this sentence and divided into two parts. The relevant section now reads:

"We assumed that everyone aged ≥ 18 years in a population of 1,000 individuals who were not pregnant or breastfeeding were screened for mf (by skin snipping). We further assumed that a set (target) number of eligible individuals consented to participate (we targeted 50 participants per cohort for the purpose of approximating the mean and variance among individual response dynamics to estimate sample sizes)."

Reviewer's comment 2.21. What is the rationale for the eligibility criterion regarding the minimum number of mf/mg skin? Is the difference between >0 and >2 relevant? Is there some clinical or epidemiological consideration here that is not being explained? As mf densities are typically highly overdispersed between individuals, wouldn't selecting densities of e.g. >10 mf/mg skin be feasible as well, and might this not make a much more noticeable difference for the required sample size?

Authors' response 2.21. We thank the Reviewer for this comment. It is common for clinical trials of antifilarial therapies to use a minimum infection intensity (i.e. minimum number of microfilariae in this case) as an eligibility criterion (see for example Awadzi et al. *PLoS Negl Trop Dis* 2014 26, e2953 <https://www.clinicaltrials.gov/ct2/show/NCT00300768>; King et al. *N Engl J Med* 2018, 8, 1801-1810 <https://clinicaltrials.gov/ct2/show/NCT01975441>; Opoku et al. *Lancet* 2018, 392, 1207-1216 <https://clinicaltrials.gov/ct2/show/NCT00790998>). However, rarely is any justification given for this choice. Hence, we varied this parameter in our simulations to explore its effect on the efficiency of the trial (e.g. the impact it has on required sample sizes). We have explained this rationale in the revised manuscript as follows:

"The CTS approach permits exploration of how decisions on minimum infection inclusion criterion and the number of skin snips used to detect mf (**Table 1** and **Figure 1**) also influence the required sample size. For examples of trials employing different design decisions see Awadzi et al.,⁴⁶ Batsa Debrah et al.,⁴⁷ Opoku et al.,⁴⁸ and Turner et al.⁴⁹"

We agree with the Reviewer's suggestion that raising the minimum number of mf would make the impact of this design choice more explicit. Hence, in the revised manuscript, we use criteria of >4 mf/mg and >8 mf/mg. Indeed, we are particularly grateful to the Reviewer for this suggestion as it revealed a subtlety that was not apparent using the previous > 2 mf/mg criterion. Namely, that the use of a minimum level of infection inclusion criterion has opposing effects on sample sizes calculated using either microfilarial intensity or prevalence. We have highlighted this in the revised manuscript as follows:

"...an increasing minimum microfilarial intensity inclusion criterion (ranging from > 0 mf/mg to > 8 mf/mg) opposing directional effects on sample sizes for the different outcome measures are evident. For microfilarial intensity, increasing the infection level inclusion criterion decreases required sample sizes because inter-participant variability in microfilarial counts is reduced (by selecting individuals with somewhat more similar microfilarial loads). By contrast, for microfilarial prevalence, increasing the infection level inclusion criterion increases the sample size. This is because the selection of more heavily infected participants decreases the number of participants who are ostensibly 'cured' (i.e. achieving zero mf) following treatment (note that macrofilaricidal efficacy is defined probabilistically as the chance that an adult *Onchocerca volvulus* worm is killed by treatment)."

We also emphasise in our discussion that there is a balance to strike between using a higher minimum infection level as an inclusion criterion to reduce sample sizes and the increased difficulty of

finding sufficient numbers of eligible participants that this entails. The relevant section of the revised manuscript now reads:

“For example, the use of a high minimum infection inclusion criterion will make it harder to find sufficient numbers of eligible participants (see Supplementary Information, Figure S4) within single communities or transmission foci with similar intensities of transmission (driven by local density of vectors). Hence, if choosing microfilarial intensity as the desired outcome measure, there is a trade-off between the benefit of reducing variability in treatment responses (and lower sample sizes) by selecting participants with more similar (and higher) infection intensities and the challenge of recruiting sufficient numbers.”

Reviewer’s comment 2.22. The authors explain that they simulate 50-person cohorts per trial arm, but from figure 5 it seems they simulated many more trial arm sizes.

Authors’ response 2.22. We used 50-participants per treatment cohort to (a) illustrate the response dynamics following treatment with a macrofilaricide or ivermectin (see Figure 3), and (b) estimate the mean and variance (standard deviation) in responses to enable approximation of the required sample sizes. Briefly, from each of 1,000 repeat simulations (for each parameter combination; originally 24,000 simulations in total, now 72,000 in total **Authors’ response 1.7** and **Authors’ response 2.21**), we estimated the mean difference in responses between groups and the standard deviation of responses within each group, which were, in turn, used to calculate (1,000 repeats of) Welch’s t -statistic. Because Welch’s t -statistic follows a non-central t -distribution, we were then able to calculate sample sizes required to achieve an 80% probability (power) of identifying a superior response in the in macrofilaricide-treated test participants compared to ivermectin-treated controls (as shown in Figure 5). This avoided having to simulate explicitly numerous cohort sizes to calculate required sample sizes which would be computationally prohibitive. Two assumptions are crucial here. First, is that our 1,000 repeats of Welch’s t -statistic (from a 50-person cohort) are adequately described by a non-central t -distribution. We confirmed this by comparing our simulated values with the (analytical) probability density function of the non-central t -distribution (see Supplementary Information, Figure S5–Figure S7). Second, is that the trial itself has a limited impact on the level of community-wide transmission (and therefore rates of reinfection). This is because our approximations rely on estimates of the mean (difference) and (within group) variability of response dynamics being consistent among trials using different sample sizes. In addition to clarifying the approach in the Methods section of the revised manuscript (for the revised text see **Authors’ response 1.8**), we have also made more explicit the underpinning assumptions:

“We further assumed that a set (target) number of eligible individuals consented to participate (we targeted 50 participants per cohort for the purpose of approximating the mean and variance among individual response dynamics to estimate sample sizes).”

“Our approach to estimating sample sizes assumes that the simulated target of 50 participants per cohort provide an adequate approximation to the ‘true’ average response and (stochastic) inter-participant variability and that these quantities (mean and inter-participant variance) are consistent across changing sample sizes. In essence, this assumption rests on the trial itself having limited immediate impact on community-wide transmission (and therefore rates of reinfection).”

Reviewer’s comment 2.23. Regarding the description of statistical test(s) used to determine whether there was significant difference between groups in the simulation: did the authors in fact perform a T-test assuming unequal variances, or is this approach starting from summary statistics (difference between group means, and group-specific standard deviations) an approximation? And if so, why this approximate approach instead of the T-test itself? And were mf densities transformed (e.g. logarithmically) before group comparison to resolve non-normality of what I imagine to be highly skewed responses? Also, (why) is this approach adequate for comparing differences in mf prevalence between groups? Wouldn’t a Fisher’s exact test be more appropriate?

Authors’ response 2.23. Our power calculations are based on an approximation to the (non-central) t -distribution (Welch’s t -test; also called an unequal variance t -test). Specifically, we used our 1,000

repeat simulations to estimate the difference in the mean responses between (treated and control) groups and the standard deviation of the responses within each group and we used these quantities to parameterize a non-central t -distribution. This is an approximate approach because strictly the sampling distribution of the difference in mean responses (i.e. the distribution of the difference in means calculated from each repeat simulation) exactly follows a t -distribution only if either the distribution of the data (within each group) is Gaussian, or the sample size is infinite (by the central limit theorem, CLT). In practice, and despite the skewed distribution of microfilarial counts (which we did not transform), we found that calculating Welch's t -statistic from a sample of 50 (or fewer) participants per treatment group provided an adequate approximation to the (non-central) t -distribution (see Supplementary Information, Figure S5–Figure S7). We also found that the approximation was adequate when applied to the difference in prevalence between groups, again because the cohort size meant that the sampling distribution of the differences was approximately normal (by the CLT, i.e. normal approximation to a binomial distribution). The adequacy of these approximations validated the approach (see **Reviewer's comment 1.10**) and permitted calculation of sample sizes without having to run a prohibitive number of simulations for different treatment cohort sizes. We could have applied a similar approach using Student's t -test but we chose Welch's t -test because it is known to be more reliable when the data being compared have unequal variances (which is the case here). The revised manuscript contains an overview of the approach in the main text (see **Authors' response 2.22**) in addition to a more detailed description (and accompanying figure) in the Supplementary Information:

“We use $\widehat{D}(\tau)$ to denote generically the mean difference between measures of either intensity (mf per mg of skin) or prevalence (percent positive for mf) in the macrofilaricide-treated, test, and ivermectin-treated, control, groups from each simulation. That is, $\widehat{D}(\tau)$ is the mean response in the control group subtracted from the mean response in the macrofilaricide-treated test group for each of the 1,000 simulations run for each parameter combination (see **Table 1** in the main text). We further denote $\widehat{\sigma}_T(\tau)$ and $\widehat{\sigma}_C(\tau)$ (T = test, C = control) as the standard deviation among responses from the 50-participant test and control cohorts included in each simulation. By invoking the central limit theorem, we assume that the distribution of $\widehat{D}(\tau)$ is approximately Gaussian such that Welch's t statistic²⁵

$$T(\tau) = \frac{\widehat{D}(\tau)}{\widehat{\sigma}(\tau)} \quad (\text{S26})$$

follows a non-central t -distribution, where $\widehat{\sigma}(\tau)$ is the approximate standard deviation among responses from both test and control groups and is estimated by,

$$\widehat{\sigma}(\tau) = \sqrt{\frac{\widehat{\sigma}_T^2(\tau)}{N_T(\tau)} + \frac{\widehat{\sigma}_C^2(\tau)}{N_C(\tau)}}. \quad (\text{S27})$$

The degrees of freedom ν associated with $T(\tau)$ is approximated by the Welch-Satterthwaite equation

$$\nu \approx \widehat{\sigma}_T^4(\tau) \left(\frac{\widehat{\sigma}_T^4(\tau)}{N_T^2(\tau)\nu_T} + \frac{\widehat{\sigma}_C^4(\tau)}{N_C^2(\tau)\nu_C} \right)^{-1}, \quad (\text{S28})$$

where $\nu_T = N_T(\tau) - 1$ and $\nu_C = N_C(\tau) - 1$. We confirmed that the distribution of $T(\tau)$ was adequately approximated by replacing $\widehat{D}(\tau)$, $\widehat{\sigma}_T(\tau)$, $\widehat{\sigma}_C(\tau)$ with their ‘true’ values $D(\tau)$, $\sigma_T(\tau)$ and $\sigma_C(\tau)$ approximated from the 1,000 repeated simulations. Hence, the probability (power, β) that $T(\tau)$ exceeds the critical value of the t

distribution t_α , where $1 - \alpha = 0.05$ is the (one-sided) probability of rejecting the null hypothesis that $D(\tau) > 0$ (type I error), is given by

$$\beta = 1 - G(t_\alpha), \tag{S29}$$

where $G(t_\alpha)$ is the cumulative distribution function of the non-central t -distribution with non-centrality parameter $D(\tau)$ and degrees of freedom ν . Assuming equal starting cohort sizes $N(0) = N_T(0) = N_C(0)$ and 10% drop out per year, we used Equation (S29) to find the smallest integer $N(0)$ that ensured a power of at least 80%, $\beta = 0.8$."

Reviewers' Comments:

Reviewer #2:

Remarks to the Author:

The authors have provided clear answers to all of my points. I'm happy to hear that they also used this opportunity to elaborate the analysis with respect to different cut-offs for the mf-density inclusion criterion, which turned out to have an interesting differential effect on power of trials relying on mf prevalence vs. mf intensity as an outcome. I have no further concerns and recommend acceptance for publication.

Luc E. Coffeng, MD PhD

Department of Public Health, Erasmus MC, University Medical Center Rotterdam, The Netherlands

Reviewer #3:

Remarks to the Author:

I feel that the authors have adequately addressed most of the comments raised by Reviewer 1, except those related to comments 1.7 to 1.9. I agree with Reviewer 1's assessment that "it is the combination of a realistic transmission model coupled with the simulations of a trial design (and its features) that offer this benefit". That is, the usefulness of the proposed method depends on the transmission model being "realistic". It has two implications: 1) The assumed values of model parameters and the transmission model itself need to be rigorously verified to ensure that they adequately reflect reality; 2) The sensitivity of trial design choices to assumed values of different parameters needs to be evaluated. For example, if the true prevalence level is 41% instead of 40% (as in comment 1.7), how would this mis-specification affect the trial design?

Reviewer #2 (Luc E Coffeng)

Remarks to the Authors. The authors have provided clear answers to all of my points. I'm happy to hear that they also used this opportunity to elaborate the analysis with respect to different cut-offs for the mf-density **inclusion** criterion, which turned out to have an interesting differential effect on power of trials relying on mf prevalence vs. mf intensity as an outcome. I have no further concerns and recommend acceptance for publication.

Authors' response. We thank the Reviewer for taking the time to review the revised manuscript and particularly for his extremely helpful suggestions on the original version. We completely agree that the additional analysis incorporated in response to the original comments has significantly improved the manuscript.

Reviewer #3

Remarks to the Authors. I feel that the authors have adequately addressed most of the comments raised by Reviewer 1, except those related to comments 1.7 to 1.9. I agree with Reviewer 1's assessment that "it is the combination of a realistic transmission model coupled with the simulations of a trial design (and its features) that offer this benefit". That is, the usefulness of the proposed method depends on the transmission model being "realistic". It has two implications: 1) The assumed values of model parameters and the transmission model itself need to be rigorously verified to ensure that they adequately reflect reality; 2) The sensitivity of trial design choices to assumed values of different parameters needs to be evaluated. For example, if the true prevalence level is 41% instead of 40% (as in comment 1.7), how would this mis-specification affect the trial design?

Authors' response. We thank the Reviewer for these comments. We completely agree with the Reviewer's assertion that the utility of the method (clinical trial simulation) is dependent on the realism offered by the transmission model and the flexibility to incorporate (model) different trial design choices. The core transmission model is an individual-based stochastic analogue of a previously developed deterministic model (for a recent review, see Basáñez et al. *Adv Parasitol* 2016, 94, 247-341) and has been shown to reflect the epidemiology and transmission dynamics of onchocerciasis across the observed range of endemicity settings, from hypoendemicity (skin microfilarial prevalence < 40%) to high hyper- or holoendemicity (skin microfilarial prevalence > 60%) (Hamley et al. *PLoS Negl Trop Dis* 2019, 13, e0007557). Moreover, because the model (EPIONCHO-IBM) is an analogue of simpler deterministic precursors, its structure is grounded in evidence generated over many years (Basáñez et al. *Phil Trans R Soc Lond B* 1999, 354, 809-826; Filipe et al. *PNAS* 2005, 102, 15265-15270; Basáñez et al. *Lancet Infect Dis* 2008, 8, 310-322; Walker et al. *Epidemics* 2017, 18, 4-15). We are thus confident that EPIONCHO-IBM reflects adequately the epidemiology and transmission dynamics of onchocerciasis. Indeed, through our involvement with the Neglected Tropical Diseases Modelling Consortium, we are using the model to inform the decisions made by global policy makers on strategies to reach the World Health Organization 2030 elimination goals for onchocerciasis (NTD Modelling Onchocerciasis Group. *Gates Open Res* 2019, 3, 1545; Hamley et al. *J Infect Dis* 2019, 20, S1-S9). We have amended the manuscript to make it more explicit that the underlying model is well-established, with a long history of development. We also include additional references to our previous work. The relevant section of the revised manuscript now reads:

“We use an adaptation of our new individual-based onchocerciasis transmission model, EPIONCHO-IBM^{45,46} (a stochastic analogue of the well-established EPIONCHO transmission model),⁴⁷⁻⁴⁹...”

In response to the original comments of Reviewer #1 (**Reviewer's comment 1.7 and 1.9**), we addressed explicitly the suggestion to explore the impact of conducting trials in different endemic foci on design choices. We had initially simulated only for a focus with an endemic microfilarial prevalence of ~ 40% (note this is approximate because of the stochastic nature of the model and the finite population host size). However, following the suggestion of Reviewer #1, we conducted additional simulations for a focus with an endemic prevalence of ~ 50% (which essentially corresponds to a sensitivity analysis on the blackfly vector biting rate, which largely drives the endemic level of infection). As is explained in the manuscript (with full details given in the Supplementary Information),

this has limited impact on the treatment response dynamics or on the identification of opportune follow-up timeframes, albeit increasing transmission (endemicity) enhances the contrasting effects on required sample sizes when using either microfilarial intensity or prevalence (presence/absence) as the primary trial outcome measure. Details of these revisions to the manuscript are given in **Authors' response 1.7** and **1.9** of Revision 1, which we copy here to facilitate inspection by the reviewer:

Reviewer's comment 1.7. Why is prevalence fixed at 40% and why do the simulations do not assess the effect of changing this assumption? I presume this should impact design choices too.

Authors' response 1.7. We had originally fixed the baseline microfilarial prevalence to 40% to reflect that the majority of ivermectin-naïve communities with greater than 40% prevalence have been under mass drug administration for many (but variable) years. We explain in the Discussion of the revised manuscript that finding ivermectin-naïve communities for the purposes of clinical trials is desirable because of the uncertain effects of multiple prior rounds of ivermectin on *Onchocerca volvulus* in those situations in which ivermectin MDA has been implemented and the difficulty with measuring accurately how many prior treatments individuals have received. (Although, with caution, the simulator can be used to predict response dynamics in communities with a prior history of MDA; this is illustrated in the README.html file that accompanies the simulator code, see also **Authors' 2.17** for further details.) Notwithstanding these considerations, and as suggested by the Reviewer, we have also included in the revised manuscript, simulations conducted with a baseline microfilarial prevalence of 50%. Realistically, it is unlikely that there exist many treatment-naïve transmission settings with greater than 50% microfilarial prevalence (because priority ivermectin MDA has focused on meso- and hyperendemic communities). Relevant sections of the revised manuscript—including reference to new figures and text included in the Supplementary Information—now read:

“We focus on trials conducted in previously ivermectin-naïve, mesoendemic transmission foci with a microfilarial prevalence among individuals aged ≥ 5 years ranging from 40% to 50% (transmission foci with higher microfilarial prevalence are likely to have been undergoing MDA for many years).”

“We developed a CTS from the individual-based onchocerciasis transmission model EPIONCHO-IBM⁴⁵ to simulate a trial conducted in an ivermectin-naïve mesoendemic community with a microfilarial prevalence in those aged ≥ 5 years of either 40% or 50%.”

“There is negligible qualitative difference in the treatment response dynamics between the lower and upper end of the mesoendemicity setting (compare **Figure 3** with **Figure S1** in the Supplementary Information).”

“Inference on opportune follow-up timeframes is qualitatively very similar between the lower and the upper end of the mesoendemicity setting, albeit there is noticeably greater variation in the difference between the microfilarial intensity outcome measure at the upper endemicity end of the setting (compare **Figure S2a** and **S2b** with **Figure 4a** and **4b**).”

“By contrast, for microfilarial prevalence, increasing the infection level inclusion criterion increases the sample size. This is because the selection of more heavily infected participants decreases the number of participants who are ostensibly ‘cured’ (i.e. achieving zero mf) following treatment (note that macrofilaricidal efficacy is defined probabilistically as the chance that an adult *Onchocerca volvulus* worm is killed by treatment). These contrasting effects are enhanced in the upper-end mesoendemic setting because participants tend to be more heavily infected (Supplementary Information, Figure S3).”

Reviewer's comment 1.9. The authors say as one of their conclusions in the discussion: “The high variability in estimated sample sizes required to detect superiority [...] reflects the potential resource-saving benefits and efficacy gains that model-based trial design can offer.” I see two caveats to these

conclusions. First, is it model based or it is the model-based simulated trial design approach that would offer these advantages. I think it is the combination of a realistic transmission model coupled with the simulations of a trial design (and its features) that offer this benefit. So, it goes beyond the modelling. Second, would this high variability also suggest that traditional fixed designs are too risky in this context as if we get the assumptions wrong a trial can go very easily from being feasible to not being so. After all, most of these are assumptions on which there will be little good evidence before the trial start. Would it not be fair that these results suggest that an Adaptive approach to trial design (where assumptions may be assessed at interim points and decisions on the trial could be made) would potentially be the approach that could materialise such potential gains shown by this work?

Authors' response 1.9 We wholly agree with the Reviewer that there are two fundamental strengths of using a simulator to inform the design of a trial. First, that the underlying transmission dynamics model can capture explicitly the effects of reinfection in disease-endemic communities. Second, that the simulator (here an individual-based transmission model) can capture various aspects of the trial design as parameters that can be varied to identify favourable choices (e.g. appropriate follow-up times, infection eligibility criteria and parasitological sampling protocols). We have emphasised these key benefits in the revised version of the manuscript:

“The transmission model that underpins the CTS accounts for the inevitable reinfection of participants by drug-naïve parasites in endemic settings. Crucially, this permits *a priori* identification of opportune follow-up times after treatment when the balance between measurable drug effects and reinfection is most favourable. Moreover, the individual-based structure of the transmission model permits explicit simulation (and exploration) of other proposed protocols, such as the use of minimum infection level eligibility criteria and sampling procedures (e.g. parasitological sampling by skin snips). Explicit simulation of transmission and trial protocols represent the key benefits of using a CTS to inform trial design.”

“The variability in estimated sample sizes required to detect superiority of a macrofilaricide compared to ivermectin reflects the potential resource-saving benefits and efficacy gains that clinical trial simulation can offer.”

We also completely agree with the Reviewer that our results suggest that adaptive approaches to trial designs could be extremely beneficial, especially for mid-trial refinements to the trial design and for making early ‘go/no go’ decisions. For example, sampling participants at interim time points—*before* the initially-proposed most opportune follow-up time—would permit the clinical trial simulator to be fitted to the data to achieve an early indication/estimate of macrofilaricidal activity and accompanying microfilaricidal activity. This information could be used to: (a) decide whether to continue or not with the trial (e.g. if insufficient macrofilaricidal activity is detected) and (b) refine the choice of follow-up time for the principal outcome measure. We had alluded to the use of adaptive approaches in the original version of the manuscript but we have strengthened and make more explicit our discussion of this important issue in the revised version:

“An alternative approach to earlier identification of macrofilaricidal activity is to collect interim parasitological (microfilarial) data before the indicated optimal follow-up time. While these data may not indicate superiority compared to ivermectin, model-based interpretation of the data (for example by fitting the CTS or related transmission dynamics models)^{8,55} will give an indication of the underlying activity and efficacy of the trialled drug. This may give important early indications to support ‘go/no go’ decisions and/or inform refinement or adaption of the trial design. For example, measuring mf from patients at interim time points—*before* the initially-proposed most opportune follow-up time—would permit the CTS to be fitted to the data to achieve an early indication/estimate of macrofilaricidal activity and accompanying microfilaricidal activity. Decisions could then be made on whether to continue the trial (e.g. depending on how the early estimate of macrofilaricidal activity compares to the desired TPP) and when the final follow up should be made (e.g. depending on the indicated level of microfilaricidal activity). The efficiency gains associated with such adaptive (as opposed to ‘fixed’) trial designs is well

recognised and has led to their increased use across the clinical medicine domain.^{71,72} Collection of interim longitudinal repeated measurements also confers added statistical power to post hoc analyses and can supplement the final comparison of trialled treatments.”